# ALPINE: Unveiling The Planning Capability of Autoregressive Learning in Language Models

**Siwei Wang**[1][*] **Yifei Shen**[1][*] **Shi Feng**[2] **Haoran Sun**[3] **Shang-Hua Teng**[4][†] **Wei Chen**[1][✉]

[1]Microsoft Research Asia (`{siweiwang, yifeishen, weic}@microsoft.com`)
[2]Harvard University (`shifeng@fas.harvard.edu`)
[3]Peking University (`sunhaoran0301@stu.pku.edu.cn`)
[4]University of Southern California (`shanghua@usc.edu`)

## Abstract

Planning is a crucial element of both human intelligence and contemporary large language models (LLMs). In this paper, we initiate a theoretical investigation into the emergence of planning capabilities in Transformer-based LLMs via their next-word prediction mechanisms. We model planning as a network path-finding task, where the objective is to generate a valid path from a specified source node to a designated target node. Our mathematical characterization shows that Transformer architectures can execute path-finding by embedding the adjacency and reachability matrices within their weights. Furthermore, our theoretical analysis of gradient-based learning dynamics reveals that LLMs can learn both the adjacency and a limited form of the reachability matrices. These theoretical insights are then validated through experiments, which demonstrate that Transformer architectures indeed learn the adjacency and an incomplete reachability matrices, consistent with our theoretical predictions. When applying our methodology to the real-world planning benchmark Blocksworld, our observations remain consistent. Additionally, our analyses uncover a fundamental limitation of current Transformer architectures in path-finding: *these architectures cannot identify reachability relationships through transitivity, which leads to failures in generating paths when concatenation is required*. These findings provide new insights into how the internal mechanisms of autoregressive learning facilitate intelligent planning and deepen our understanding of how future LLMs might achieve more advanced and general planning-and-reasoning capabilities across diverse applications.

## 1 Introduction

Large language models (LLMs) such as ChatGPT have impressed many with their powerful capabilities across a wide range of tasks, including language processing, knowledge extraction, reasoning, planning, coding, tool use, and more [1]. However, we continue to be intrigued by the underlying mechanisms that fuel the power of LLMs. While all current LLMs are built on the Transformer architecture, which uses autoregressive learning to predict the next word in a language sequence, the overarching question remains:

*Why does the process of next-word prediction give rise to intelligence?*

There is no definite answer to this question yet, but researchers are approaching the problem from various angles, aiming to characterize the power and limitations of LLMs, as well as to capture their underlying acquisition, abstraction, generalization, adaptation, and reasoning mechanisms.

---

[*] denotes equal contributions. Corresponding author: Wei Chen (`weic@microsoft.com`)
[†]Supported by a Simons Foundation Investigator Award.

38th Conference on Neural Information Processing Systems (NeurIPS 2024).

Recently, the mechanisms of grammar learning, knowledge manipulation, scaling laws, and arithmetic operations have been empirically uncovered [4, 3, 5, 2, 31, 11]. Furthermore, theoretical analyses have been conducted on in-context learning, chain-of-thought, and other forms of reasoning [30, 8, 7, 27]. Beyond these, LLMs' capability for planning—a fundamental component of human intelligence—has also drawn considerable attention. Planning is involved in nearly every aspect of our daily life, such as organizing a task at work, planning a trip, or seeking a mathematical proof of a theorem. Additionally, task planning plays a pivotal role in state-of-the-art LLM-empowered autonomous agents, such as HuggingGPT [18], Voyager [23], and Reflection [19]. Understanding how LLMs complete a planning task can shed light on how the seemingly low-level statistical task of next-word prediction transforms into a high-level intelligent process. Several prior studies have empirically evaluated the planning capabilities of LLMs, yielding both positive and negative results [16, 22]. However, the current results are incomplete and do not fully explain why LLMs can or cannot successfully accomplish specific planning tasks (see Appendix A for a detailed discussion of related works).

Given that planning often involves making sequential selections of next steps to achieve a desired goal, it naturally relates to the path-finding task in networks. For example, autonomous agents (e.g., HuggingGPT [18]) for scheduling API calls can be likened to finding a call path in the API call graph; a mathematical proof can be seen as a proof path from the axioms to the final theorem [21]; and a step-by-step solution to a grade-school math problem can be viewed as a path in the dependency graph among the variables [28, 29]. Many previous studies on LLM planning capabilities are related to path-finding. e.g., an LLM planning benchmark called Blocksworld [22] can be viewed as path-finding from the initial state of the blocks to the final state in a state transition graph. Furthermore, in neuroscience, planning is often evaluated through path-finding in a maze [26]. Consequently, in this paper, we abstract planning in LLM learning as the following path-finding task: given an underlying directed graph, a Transformer architecture is provided with training data consisting of a collection of paths that specify the source node, the target node, and a path from the source to the target. The task of the language model is then to generate a path for a new source-target pair. In addition to measuring the performance of the trained model, we examine the internal weighting mechanism and the learning dynamics of the Transformer architecture during the learning and problem-solving process. This research is part of our broader project, ALPINE (Autoregressive Learning for Planning In NEtworks), which aims to answer the overarching question on the connection between the process of next-word prediction and the emergence of intelligence through the lens of planning.

**Our Contributions**: Our project initiates a theoretical investigation into the development of planning capabilities in Transformer-based language models by focusing on characterizing their expressiveness and learning dynamics in the path-finding task. First, in Theorem 2, we present a mathematical construction of a Transformer that encodes both the adjacency and reachability matrices of the network, thereby establishing that Transformer architectures possess the expressive capacity to complete the path-finding task. Then, in Theorem 3, we prove that when applying gradient descent to minimize the cross-entropy loss on the training data, a model based on a simplified Transformer architecture can extract the adjacency matrix and a limited form of the reachability matrix, using them to mimic human-like intelligence in path-finding. Our theoretical analysis further reveals a fundamental limitation of current Transformer architectures: they do not learn certain types of reachability, particularly transitive reachability, resulting in an incomplete ability to reason about future steps when planning. To validate our theoretical findings, we conduct extensive experiments training Transformers on the path language using autoregressive learning. First, these experiments demonstrate that Transformers achieve high accuracy in the path-finding task (Figure 3). Second, we show that it is indeed possible to extract both the adjacency and a limited form of the reachability matrices from the Transformers' weights (Figures 1,2,5,6(a)). Third, we observe a significant drop in test accuracy when the source and target nodes are connected only through concatenated path segments in the training data (Figure 6). These findings align with our theoretical analysis, confirming that *current Transformers have limitations in learning transitive reachability relationships, unlike human intelligence*. Finally, we validate these results on a real-world task planning benchmark, Blocksworld [22], which directly corresponds to the path-finding problem (see Appendix F).

## 2 Preliminaries

Throughout this paper, we use the following notations for matrices and vectors: $\boldsymbol{a}$ and $\boldsymbol{A}$ stand for a column vector and a matrix, respectively. Notations $\boldsymbol{a}_{(i)}$ and $\boldsymbol{A}_{(i,j)}$ denote the $i^{th}$ entry of vector $\boldsymbol{a}$ and the $(i,j)^{th}$ entry in matrix $\boldsymbol{A}$, respectively. We also denote the $i^{th}$ row of matrix $\boldsymbol{A}$ by $\boldsymbol{A}_{(i,:)}$.

## 2.1 Autoregressive Transformer Architecture and Loss Function

In this paper, we adopt the standard GPT architecture [17]. We use the following notation for the architecture and loss function in our analysis. Let $N$ denote the sequence length, $d$ the embedding size, $H$ the number of heads, $d_k = d/H$ the embedding size per head, and $M$ the vocabulary size. One key component of the architecture is the attention mechanism, which is formulated as:

$$\text{Attention}(\boldsymbol{Q}, \boldsymbol{K}, \boldsymbol{V}) = \textbf{softmax}\left(\frac{\boldsymbol{Q}\boldsymbol{K}^{\top}}{\sqrt{d_k}}\right)\boldsymbol{V} \tag{1}$$

where $\boldsymbol{Q} \in \mathbb{R}^{N \times d_k}$, $\boldsymbol{K} \in \mathbb{R}^{N \times d_k}$, $\boldsymbol{V} \in \mathbb{R}^{N \times d_k}$ are the query, key, and value matrices, respectively. Denoting $\boldsymbol{X} \in \mathbb{R}^{N \times d}$ as input, the multi-head attention is computed as $\text{MHA}(\boldsymbol{X}) = \text{Concat}_{i \in [H]}(\text{Attention}(\boldsymbol{X}\boldsymbol{W}_i^Q, \boldsymbol{X}\boldsymbol{W}_i^K, \boldsymbol{X}\boldsymbol{W}_i^V))$, where $\boldsymbol{W}_i^Q \in \mathbb{R}^{d \times d_k}$, $\boldsymbol{W}_i^K \in \mathbb{R}^{d \times d_k}$, $\boldsymbol{W}_i^V \in \mathbb{R}^{d \times d_k}$ are the learnable weight matrices for the query, key, and value matrices of the $i$-th head.

The feed-forward layer is a two-layer multi-layer perceptron (MLP) defined as follows:

$$\text{FFN}(\boldsymbol{X}) = \max(\boldsymbol{0}, \boldsymbol{X}\boldsymbol{W}_1 + \boldsymbol{1}_{N \times 1}\boldsymbol{b}_1^{\top})\boldsymbol{W}_2 + \boldsymbol{1}_{N \times 1}\boldsymbol{b}_2^{\top} \tag{2}$$

where $\boldsymbol{W}_1 \in \mathbb{R}^{d \times 4d}$, $\boldsymbol{W}_2 \in \mathbb{R}^{4d \times d}$, $\boldsymbol{b}_1 \in \mathbb{R}^{4d}$, and $\boldsymbol{b}_2 \in \mathbb{R}^d$ are the learnable weight matrices and $\boldsymbol{1}_{N \times x}$ is the all-one matrix with dimension $N \times x$. Finally, one Transformer layer is defined as:

$$\text{Transformer}(\boldsymbol{X}) = \text{FFN}(\text{LN}_2(\text{MHA}(\text{LN}_1(\boldsymbol{X})) + \boldsymbol{X})) + \text{MHA}(\text{LN}_1(\boldsymbol{X})) + \boldsymbol{X} \tag{3}$$

where $\text{LN}_1$ and $\text{LN}_2$ are two layer normalizations.

With these essential components in place, we proceed to introduce the procedures of GPT. The training data consists of many sequences of tokens, where each sequence is expressed as $\boldsymbol{u} = (u_1, \cdots, u_N)$, in which $u_n$ denotes the token id for the $n$-th token in sequence $\boldsymbol{u}$. We first represent the tokens in $\boldsymbol{u}$ by a one-hot embedding matrix $\boldsymbol{U} \in \mathbb{R}^{N \times M}$, where $\boldsymbol{U}_{(n, u_n)} = 1$ and 0 elsewhere. Then there are learnable token embedding matrix $\boldsymbol{W}_t \in \mathbb{R}^{M \times d}$ and positional embedding matrix $\boldsymbol{W}_p \in \mathbb{R}^{N \times d}$, and the input $\boldsymbol{H}_0 = \boldsymbol{U}\boldsymbol{W}_t + \boldsymbol{W}_p \in \mathbb{R}^{N \times d}$. This input $\boldsymbol{H}_0$ is fed into an $L$-layer Transformer, i.e., $\boldsymbol{H}_l = \text{Transformer}(\boldsymbol{H}_{l-1}) \in \mathbb{R}^{N \times d}$ for $l = 1, \cdots, L$.

Finally, the output embedding goes through another layer normalization $\text{LN}_t$, and then it is multiplied by a learnable output weight matrix $\boldsymbol{W}_o \in \mathbb{R}^{d \times M}$ to convert back to probability weights over all possible tokens. We calculate the output probability vector at position $n$, denoted by $\hat{\boldsymbol{u}}_{(n+1)}$, using $\hat{\boldsymbol{u}}_{(n+1)} = \textbf{softmax}((\text{LN}_t(\boldsymbol{H}_L))_{(n,:)}\boldsymbol{W}_o), 1 \leq n < N$. This probability vector is used to predict the token for position $n + 1$, which reflects the autoregressive learning paradigm.

The adopted loss function is the *cross-entropy loss* for the next token prediction, given by:

$$\ell(\boldsymbol{u}) = -\sum_{n=1}^{N-1}\sum_{k=1}^{M} \boldsymbol{U}_{(n+1,k)} \log \hat{\boldsymbol{u}}_{(n+1),k} \tag{4}$$

## 2.2 Path-Planning Dataset: Syntax and Data Sources

The dataset is designed to evaluate GPT's path-planning capability on simple graphs. It is generated from a directed graph $\mathcal{G} = (\mathcal{V}, \mathcal{E})$, where $\mathcal{V}$ is the set of nodes, and $\mathcal{E}$ is the set of edges. For any two nodes $u, v \in \mathcal{V}$, $(u, v) \in \mathcal{E}$ means that there is a directed edge from $u$ to $v$ in $\mathcal{G}$. A pair of source node $s$ and target node $t$ is considered a *valid pair* if $\mathcal{G}$ contains at least one path from $s$ to $t$.

The training dataset $\mathcal{D}$ contains sequences in the format "$s\ t\ s\ a\ b\ c\ t$ \n", where $s$ represents the source node token, $t$ the target node token, $s\ a\ b\ c\ t$ are tokens for nodes in a valid path from $s$ to $t$, and \n indicates the end of the sequence. During testing, we provide valid pairs of source and target nodes in the format "$s\ t$". The model is tasked with completing the remaining tokens in the sequence. The completion is considered correct if the model generates a valid path with the correct syntax.

# 3 Expressiveness and Learning Dynamics of Transformer Models

## 3.1 Expressiveness

In our path-finding task, the essential step for completing a path is to predict the next node based on the current information. It is evident that to predict the subsequent node on the path, only information related to the current node and the target node is necessary. Algorithm 1 introduces a idealized method that utilizes both the adjacency and reachability matrices of the graph. The true adjacency matrix $\boldsymbol{A}^{\text{true}}$ and the true reachability matrix $\boldsymbol{R}^{\text{true}}$ are defined as:

---
**Algorithm 1** A handcrafted path-finding algorithm

---
1: **Input:** Adjacency matrix $\boldsymbol{A}$, reachability matrix $\boldsymbol{R}$, source node $s$, target node $t$
2: Set path $P = [s\ t\ s]$ and set current node $i = s$
3: **while** $i \neq t$ **do**
4:     Obtain $S = \{k|\boldsymbol{A}_{(i,k)} = 1 \text{ and } \boldsymbol{R}_{(t,k)} = 1\}$
5:     Randomly sample next node $k$ from $S$
6:     Append $k$ to path $P$, and set $i = k$
7: **end while**
8: **output** path $P$

---

$$\boldsymbol{A}^{\text{true}}_{(i,k)} = \begin{cases} 1, & \text{if } (i,k) \in \mathcal{E}, \\ 0, & \text{otherwise.} \end{cases} \qquad \boldsymbol{R}^{\text{true}}_{(t,k)} = \begin{cases} 1, & \text{if } k \text{ can reach } t \text{ in } \mathcal{G}, \\ 0, & \text{otherwise.} \end{cases}$$

**Fact 1.** *Assuming that $t$ is reachable from $s$, then Algorithm 1 is guaranteed to output a valid path with input $\boldsymbol{A} = \boldsymbol{A}^{\text{true}}$ and $\boldsymbol{R} = \boldsymbol{R}^{\text{true}}$.*

To illustrate the expressive capacities of the Transformer model, we first demonstrate how to manually construct a Transformer that can perform the path-finding task by simulating Algorithm 1.

**Theorem 2.** *Given a graph $\mathcal{G}$ (with adjacency matrix $\boldsymbol{A}^{\text{true}}$ and reachability matrix $\boldsymbol{R}^{\text{true}}$), for every $\varepsilon > 0$, there exists a 1-layer, 1-head, and $O(|\mathcal{V}|)$-embedding-size Transformer model that generates a valid path for every valid source-target pair $(s,t)$ with probability at least $1 - \varepsilon$.*

The proof involves encoding the adjacency and reachability matrices into the weights of the FFN and attention layers, respectively, while mimicking the computation of Algorithm 1 (see Appendix B).

## 3.2 Learning Dynamics

Having established the mathematical existence of a Transformer model capable of accomplishing path-finding in a given network, as demonstrated in Theorem 2, we now shift our focus to the following fundamental question: *Can the Transformer architecture, trained on sufficient path data with an autoregressive loss as in Equation (4) and using the gradient descent (GD) method, learn the adjacency and reachability matrices and carry out path-finding similar to the idealized Algorithm 1?*

Theoretically, we notice that the Transformer may not be able to learn the true adjacency and reachability matrices for the underlying graph. Instead, it can only learn the relevant information that is directly encoded in the observed training data $\mathcal{D}$. Therefore, we define the *observed* adjacency and reachability matrices based on the training data $\mathcal{D}$ as follows.

$$\boldsymbol{A}^{\text{obs}}_{(i,k)}(\mathcal{D}) = \begin{cases} 1, & \text{if } \exists \boldsymbol{u} \in \mathcal{D}, n \in [3, N-1] \text{ s.t. } u_n = i, u_{n+1} = k \\ 0, & \text{otherwise} \end{cases}$$

$$\boldsymbol{R}^{\text{obs}}_{(t,k)}(\mathcal{D}) = \begin{cases} 1, & \text{if } \exists \boldsymbol{u} \in \mathcal{D}, n \in [4, N] \text{ s.t. } u_2 = t, u_n = k \\ 0, & \text{otherwise.} \end{cases}$$

Naturally, the observed adjacency matrix $\boldsymbol{A}^{\text{obs}}(\mathcal{D})$ only records the edges $(i,k)$ that appear in some path within the training data $\mathcal{D}$. On the other hand, the observed reachability matrix $\boldsymbol{R}^{\text{obs}}(\mathcal{D})$ exhibits more nuanced distinctions from the true reachability matrix. It only records that node $t$ is reachable from node $k$, if the training data $\mathcal{D}$ contains a path (sequence) whose target node is $t$ and $k$ appears as a non-source node on the path. We call such pairs $(t,k)$ *observed reachable pairs*. Noticeably, reachability through transitivity, i.e., through concatenation of path segments in $\mathcal{D}$, is not observed.

Here we consider the following simplified 1-layer and 1-head Transformer structure: a) The attention weight is only on the target node (the second token), i.e., we manually set every row in $\mathbf{softmax}\left(\frac{\boldsymbol{Q}\boldsymbol{K}^\top}{\sqrt{d_k}}\right)$ in Eq. (1) to be a one-hot vector with the second coordinate being 1 (this is validated in our experiments shown in Figure 4), and set the positional embedding matrix $\boldsymbol{W}_p = \boldsymbol{0}$; b) We remove all the layer normalizations, and use $\text{FFN}(\boldsymbol{X}) = \boldsymbol{X}\boldsymbol{W}^M$ instead of Eq. (2), $\text{Transformer}(\boldsymbol{X}) = \text{FFN}(\boldsymbol{X}) + \text{MHA}(\boldsymbol{X})$ instead of Eq. (3); c) The token embedding matrix $\boldsymbol{W}_t$ and the output weight matrix $\boldsymbol{W}_o$ are set to be identity. The embedding size is the same as the vocabulary size ($d = M$), and we only consider the cross-entropy loss of predicting the next node.

Since there is only one layer and one head, we use $\boldsymbol{W}^V$ to represent the weight of the value matrix in the attention layer. Under the above Transformer structure,

$$(\boldsymbol{H}_L)_{(n,:)}\boldsymbol{W}_o = (\boldsymbol{U}\boldsymbol{W}_t\boldsymbol{W}^M + \boldsymbol{\alpha}\boldsymbol{U}\boldsymbol{W}_t\boldsymbol{W}^V)_{(n,:)}\boldsymbol{W}_o = (\boldsymbol{U}\boldsymbol{W}^M + \boldsymbol{\alpha}\boldsymbol{U}\boldsymbol{W}^V)_{(n,:)} = \boldsymbol{W}^M_{(u_n,:)} + \boldsymbol{W}^V_{(u_2,:)},$$

where $\boldsymbol{\alpha}$ is the manually set attention weight matrix (every row is a one-hot vector with the second coordinate being 1). Therefore, the probability vector when predicting the $(n + 1)$-th token is $\textbf{softmax}(\boldsymbol{W}^M_{(u_n,:)} + \boldsymbol{W}^V_{(u_2,:)})$, and the prediction probability $\hat{u}_{n+1,k}$ equals

$$\hat{u}_{n+1,k} = \frac{\exp(\boldsymbol{W}^M_{(u_n,k)} + \boldsymbol{W}^V_{(u_2,k)})}{\sum_\ell \exp(\boldsymbol{W}^M_{(u_n,\ell)} + \boldsymbol{W}^V_{(u_2,\ell)})}. \tag{5}$$

Let $N_{i,j,k}$ be the number of times in $\mathcal{D}$ that: a) the current node is $i$; b) the target node is $j$; c) the next node is $k$, and let $N_{i,j} = \sum_k N_{i,j,k}$, then we have the following theorem.

**Theorem 3.** *Under the cross-entropy loss $\ell(\mathcal{D})$, for all $(i, k)$ pairs, i) if $\sum_j N_{i,j} = 0$, then $\frac{\partial\ell(\mathcal{D})}{\partial\boldsymbol{W}^M_{(i,k)}}$ is always 0; ii) if $\sum_j N_{i,j} > 0$ but $\sum_j N_{i,j,k} = 0$, then $\frac{\partial\ell(\mathcal{D})}{\partial\boldsymbol{W}^M_{(i,k)}}$ is always positive; iii) if $\sum_j N_{i,j,k} > 0$, then $\frac{\partial\ell(\mathcal{D})}{\partial\boldsymbol{W}^M_{(i,k)}}$ is negative when $\boldsymbol{W}^M_{(i,k)} \to -\infty$. Similarly, for all $(j, k)$ pairs, i) if $\sum_i N_{i,j} = 0$, then $\frac{\partial\ell(\mathcal{D})}{\partial\boldsymbol{W}^V_{(j,k)}}$ is always 0; ii) if $\sum_i N_{i,j} > 0$ but $\sum_i N_{i,j,k} = 0$, then $\frac{\partial\ell(\mathcal{D})}{\partial\boldsymbol{W}^V_{(j,k)}}$ is always positive; iii) if $\sum_i N_{i,j,k} > 0$, then $\frac{\partial\ell(\mathcal{D})}{\partial\boldsymbol{W}^V_{(j,k)}}$ is negative when $\boldsymbol{W}^V_{(j,k)} \to -\infty$.*

*Proof Sketch.* By the definition of the cross-entropy loss in Eq.(4), and the prediction probability vector in Eq.(5), the total cross-entropy loss of the model (with matrices $\boldsymbol{W}^M$, $\boldsymbol{W}^V$) is

$$\ell(\mathcal{D}) = -\sum_{i,j,k} N_{i,j,k}(\boldsymbol{W}^M_{(i,k)} + \boldsymbol{W}^V_{(j,k)}) + \sum_{i,j} N_{i,j} \log\left(\sum_\ell \exp(\boldsymbol{W}^M_{(i,\ell)} + \boldsymbol{W}^V_{(j,\ell)})\right).$$

Then we can get that: (the proof for the $\boldsymbol{W}^V$ part is similar)

$$\frac{\partial\ell(\mathcal{D})}{\partial\boldsymbol{W}^M_{(i,k)}} = -\sum_j N_{i,j,k} + \sum_j N_{i,j} \frac{\exp(\boldsymbol{W}^M_{(i,k)} + \boldsymbol{W}^V_{(j,k)})}{\sum_\ell \exp(\boldsymbol{W}^M_{(i,\ell)} + \boldsymbol{W}^V_{(j,\ell)})}. \tag{6}$$

In case i), $\sum_j N_{i,j} = 0$ implies that $\sum_j N_{i,j,k} = 0$. Hence $\frac{\partial\ell(\mathcal{D})}{\partial\boldsymbol{W}^M_{(i,k)}}$ is always 0.

In case ii), $\sum_j N_{i,j} > 0$ implies that the second term in Eq. (6) is positive, while $\sum_j N_{i,j,k} = 0$ implies that the first term in Eq. (6) is 0. Hence $\frac{\partial\ell(\mathcal{D})}{\partial\boldsymbol{W}^M_{(i,k)}}$ is always positive.

In case iii), when $\sum_j N_{i,j} > 0$ and $\boldsymbol{W}^M_{(i,k)} \to -\infty$, the second term in Eq. (6) converges to 0, and it is smaller than $\sum_j N_{i,j,k} > 0$. Hence, $\frac{\partial\ell(\mathcal{D})}{\partial\boldsymbol{W}^M_{(i,k)}}$ is negative when $\boldsymbol{W}^M_{(i,k)} \to -\infty$. □

The above technical theorem directly leads to a theoretical explanation on how the model learns the adjacency and reachability information, as explained below.

**Learning the adjacency matrix.** Let $\mathcal{E}(\mathcal{D})$ denote the set of edges appearing in the training dataset $\mathcal{D}$, which corresponds to the observed adjacency matrix $\boldsymbol{A}^{\text{obs}}(\mathcal{D})$. For any $(i, k) \in \mathcal{E}(\mathcal{D})$, $\sum_j N_{i,j,k} > 0$, and for any $(i', k') \notin \mathcal{E}(\mathcal{D})$, $\sum_j N_{i',j,k'} = 0$. Then from the above theorem, under the gradient descent learning paradigm, $\boldsymbol{W}^M_{(i',k')}$ will keep decreasing (since its gradient is always positive), while $\boldsymbol{W}^M_{(i,k)}$ will not (since its gradient becomes negative when its value is sufficiently negative). This tends to make $\boldsymbol{W}^M_{(i,k)}$ higher than $\boldsymbol{W}^M_{(i',k')}$ after training. In this way, the Transformer model *learns the information about the observed adjacency matrix* with matrix $\boldsymbol{W}^M$.

To facilitate comprehension, we conducted a simple experiment on the simplified Transformer, and present the results in Figure 1, In this experiment, we generate a 10-node graph, and use 3 different training datasets $\mathcal{D}_1, \mathcal{D}_2, \mathcal{D}_3$ based on this graph. $\mathcal{D}_1$ contains all the paths with length 1; $\mathcal{D}_2$ contains

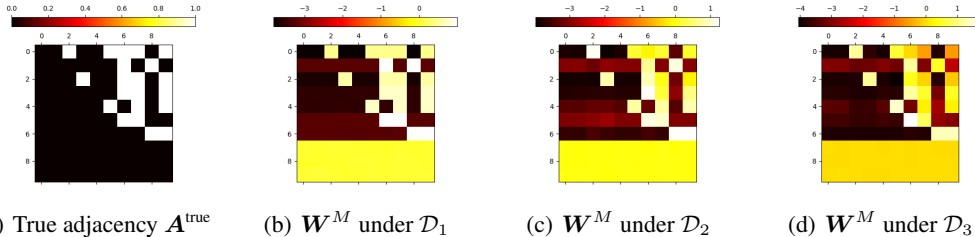

(a) True adjacency $\boldsymbol{A}^{\text{true}}$    (b) $\boldsymbol{W}^M$ under $\mathcal{D}_1$    (c) $\boldsymbol{W}^M$ under $\mathcal{D}_2$    (d) $\boldsymbol{W}^M$ under $\mathcal{D}_3$

Figure 1: Empirical verification regarding the learning of the adjacency matrix.

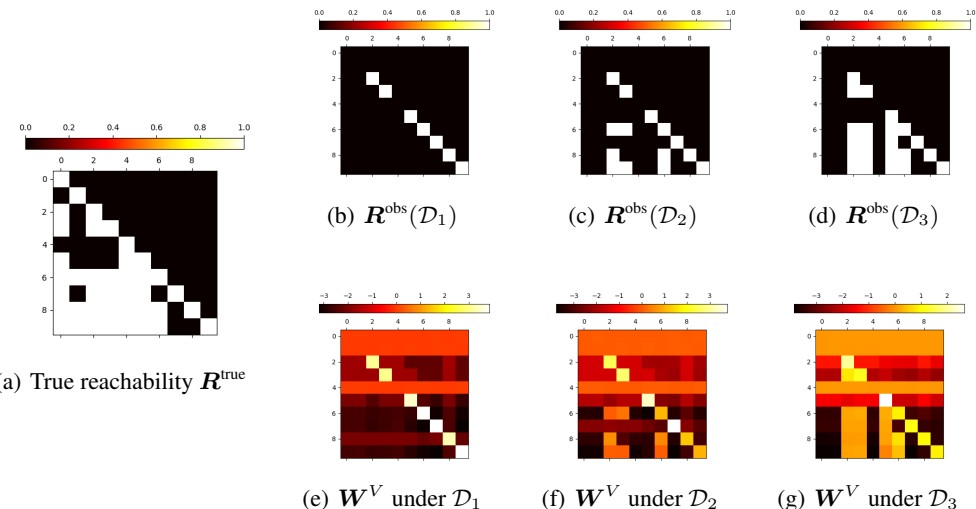

(a) True reachability $\boldsymbol{R}^{\text{true}}$

(b) $\boldsymbol{R}^{\text{obs}}(\mathcal{D}_1)$    (c) $\boldsymbol{R}^{\text{obs}}(\mathcal{D}_2)$    (d) $\boldsymbol{R}^{\text{obs}}(\mathcal{D}_3)$

(e) $\boldsymbol{W}^V$ under $\mathcal{D}_1$    (f) $\boldsymbol{W}^V$ under $\mathcal{D}_2$    (g) $\boldsymbol{W}^V$ under $\mathcal{D}_3$

Figure 2: Empirical verification regarding the learning of the observed reachability matrix.

all the paths with length 1 and 20% of the paths with length higher than 1; and $\mathcal{D}_3$ contains all the possible paths. Figure 1(a) is the true adjacency matrix of the graph, which is also the observed adjacency matrix for the three datasets. Figure 1(b), 1(c), 1(d) are the $\boldsymbol{W}^M$ matrices with training datasets $\mathcal{D}_1, \mathcal{D}_2, \mathcal{D}_3$, respectively. Upon observation, it becomes evident that these $\boldsymbol{W}^M$ matrices all successfully capture the structural information from the adjacency matrix.

**Learning the reachability matrix.** Similar to the process of learning the adjacency matrix, since only *observed reachable pairs* $(j, k)$ have $\sum_i N_{i,j,k} > 0$, the gradient descent learning paradigm tends to make the $\boldsymbol{W}^V_{(j,k)}$ terms corresponding to observed reachable pairs $(j, k)$ higher than the $\boldsymbol{W}^V_{(j',k')}$ terms corresponding to non-observed reachable pairs $(j', k')$ (which is either not reachable or not observed) after the training. In this way, the Transformer model *captures the structural information of observed reachability matrix* with weight matrix $\boldsymbol{W}^V$.

Figure 2 shows the correlation between $\boldsymbol{W}^V$ and the observed reachabilities under different dataset $\mathcal{D}$'s in the above experiment. Figure 2(a) is the real reachability matrix of the graph; Figure 2(b), 2(c), 2(d) are the observed reachability matrices in datasets $\mathcal{D}_1, \mathcal{D}_2, \mathcal{D}_3$, respectively; and Figure 2(e), 2(f), 2(g) are the $\boldsymbol{W}^V$ matrices with training datasets $\mathcal{D}_1, \mathcal{D}_2, \mathcal{D}_3$, respectively. These illustrations show that all the weight matrices $\boldsymbol{W}^V$ can satisfactorily learn the information of the observed reachabilities present in the training datasets, but cannot deduce any non-observed reachabilities.

**Predicting the next node on a path.** From Eq.(5), the probability vector for predicting the next node is $\mathbf{softmax}(\boldsymbol{W}^M_{(u_n,:)} + \boldsymbol{W}^V_{(u_2,:)})$, where $u_n$ represents the current node, and $u_2$ represents the target node. This resembles the procedure in Algorithm 1: it predicts the next node $k$ such that both $\boldsymbol{W}^M_{(u_n,k)}$ is high (corresponding to $\boldsymbol{A}_{(u_n,k)} = 1$) and $\boldsymbol{W}^V_{(u_2,k)}$ is high (corresponding to $\boldsymbol{R}_{(u_2,k)} = 1$).

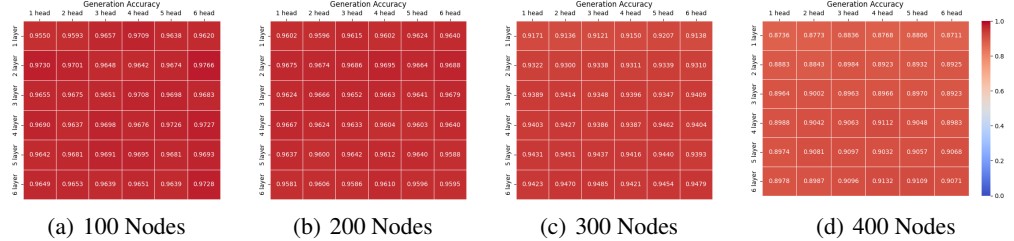

Figure 3: Accuracy on the test datasets with embedding size $d = 120$.

In summary, our theoretical analysis demonstrates that a simplified one-layer, one-head autoregressive Transformer (with perfect attention) can effectively learn crucial adjacency and reachability information from the training data through gradient descent training. Moreover, it can utilize this learned information to predict the next node akin to the decision-making process of a human algorithm designer in similar scenarios. This suggests that, when confronted with the path-finding or more general planning task with a given goal, the Transformer learns the structural information to associate the next step with both the current step and the goal, enabling it to generate the subsequent task step. Nevertheless, the Transformer's limitation in learning only the observed reachability matrix—without deducing the complete reachability matrix—hints at potential constraints on the goal-oriented information it can acquire. This limitation may result in the Transformer failing to grasp novel reachability relationships derived from the transitivity of reachability relations, unlike human intelligence.

## 4 Empirical Evaluations: Peeking into a Trained Transformer

In this section, we conduct extensive experiments on the path-finding task using the general Transformer architecture as described in Section 2.1. The datasets are generated as described below.

The DAG is generated randomly based on two parameters: the number of nodes $n$, and the probability of edge $p = 0.1$: For any $1 \le i < j \le n$, there is an edge $(i, j) \in \mathcal{E}$ with probability $p$. Given the DAG, we first find all the possible reachable pairs $(s, t)$. Then these reachable pairs are separated into the training set (w.p. 0.5) and the test set (w.p. 0.5), but if edge $(s, t) \in \mathcal{E}$, we always put $(s, t)$ in the training set. For a reachable pair $(s, t)$ in the training set, we generate $m = 20$ random paths that start at $s$ and end at $t$, and put these $m$ paths into the training dataset. When generating the random path, at each current node $i$, we find all the possible $k \in \mathcal{V}$ such that $\boldsymbol{A}^{\text{true}}_{(i,k)} = 1$ and $\boldsymbol{R}^{\text{true}}_{(t,k)} = 1$ (i.e., there is an edge $(i, k) \in \mathcal{E}$, and $k$ could also reach the target node $t$), and uniformly choose a random one in them. Moreover, if $(s, t) \in \mathcal{E}$, we always put the one-edge path "$s\ t\ s\ t\ $\n" in the training dataset to guarantee that all edges appear at least once in the training data.

### 4.1 Accuracy on Test Dataset

We train Transformer models on the aforementioned training dataset and subsequently evaluate the performance of these models using the pairs in the test dataset. The correctness of a model's output is determined based on its validity in terms of syntax and whether it corresponds to a valid path from $s$ to $t$. In our experiments, we employ Transformer models with an embedding size of $d = 120$. We conduct tests using various configurations, ranging from 1-layer and 1-head to 6-layer and 6-head, while considering different graph sizes, with number of nodes $n$ ranging from 100 to 500. The accuracy results take average over 10000 trials, and are presented in Figure 3 (due to space limits, some results are deferred to Appendix E, and all of them are consistent with our conclusions). From these results, we make the following observations: a) When comparing the figures, the accuracy tends to decrease as the number of nodes increases; b) When examining each row, the accuracy remains relatively stable even as the number of attention heads increases; c) When examining each column, the accuracy shows at most a slight improvement as the number of layers increases.

The above observations suggest that the embedding size is the most important hyperparameter that affects the accuracy of the model. On the one hand, when the embedding size is sufficiently large compared to the graph size, even 1-layer and 1-head models perform well. This coincides with our theoretical analysis, which shows that when the embedding size equals to the graph size, the 1-layer and 1-head structure is enough to predict the next nodes accurately. On the other hand, when the

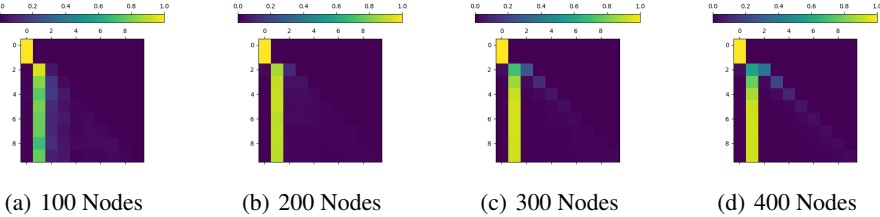

(a) 100 Nodes      (b) 200 Nodes      (c) 300 Nodes      (d) 400 Nodes

Figure 4: The average attention in the 1-layer and 1-head Transformers.

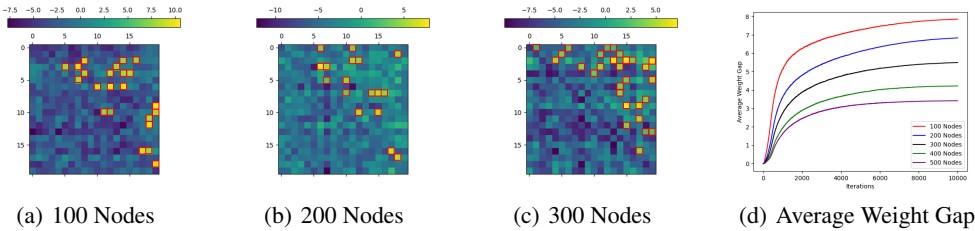

(a) 100 Nodes      (b) 200 Nodes      (c) 300 Nodes      (d) Average Weight Gap

Figure 5: The first 20 rows and columns of $\boldsymbol{W}^{M'}$ (the red boxes correspond to 1's in the adjacency matrix), and the average weight gap between edge terms and non-edge terms in $\boldsymbol{W}^{M'}$.

embedding size is small compared to the graph size, even 6-layer and 6-head Transformers cannot achieve good performance. Because of this, in the following, we concentrate on the explainability of the 1-layer and 1-head Transformer models.

## 4.2 Peeking into a Trained Transformer

**Attention.** In our analysis, we assume that the attention is fixed on the target node. Is this true for the Transformer models learned from real data? The corresponding results are shown in Figure 4. These results are obtained by looking into the attention mechanism of the 1-layer and 1-head Transformer models, and showing the average (taking on the test dataset) matrix of $\textbf{softmax}\left(\frac{\boldsymbol{QK}^\top}{\sqrt{d_k}}\right)$, of which the $n$-th row represents the attention vector for predicting the $(n+1)$-th token.

Note that the second column in these figures represents the attention weights on the second token, which corresponds to the target node in our test data. We can see that, when predicting the next tokens, almost all the attention weights are concentrated on this column, especially for those models with higher accuracy (Figure 4(a) for $n = 100$ and Figure 4(b) for $n = 200$). This demonstrates that indeed the Transformer model learns the correct attention for the path-finding task, and our assumption on the attention weights for the theoretical analysis is reasonable.

**Adjacency Matrix.** In the 1-layer and 1-head Transformers, let $\boldsymbol{W}^{M'}$ (shown in Figure 5) be the matrix whose $i$-th row is FFN $\left(\boldsymbol{e}_i^\top \boldsymbol{W}_t\right) \boldsymbol{W}_o + (\boldsymbol{e}_i^\top \boldsymbol{W}_t)\boldsymbol{W}_o$, where $\boldsymbol{e}_i$ is the one-hot column vector that represents the token for node $i$. Based on the Transformer computation, intuitively this matrix is one of the components in the output that contains information related to the current node. The detailed reason for choosing this matrix is explained in Appendix D.

In Figure 5(a), the $\boldsymbol{W}^{M'}$ matrix and the adjacency matrix are highly aligned: all large entries in the $\boldsymbol{W}^{M'}$ matrix correspond to real edges, and all real edges correspond to large entries in the $\boldsymbol{W}^{M'}$ matrix. This high accuracy is because the embedding size $d = 120$ is higher than the number of nodes $n = 100$. If the embedding size is lower than the graph size (Figures 5(b), 5(c)), we inevitably lose some accuracy when approximating the adjacency matrix by the product of matrices with rank smaller than the graph size. Even so, there is still high relevance between $\boldsymbol{W}^{M'}$ and the adjacency matrix: almost all real edges correspond to large entries in the $\boldsymbol{W}^{M'}$ matrix.

In Figure 5(d), we show the gap between the average weight corresponding to edges (i.e., the average of $\boldsymbol{W}^{M'}_{(i,j)}$'s with $i < j$ and $(i,j) \in \mathcal{E}$) and the average weight corresponding to non-edges (i.e.,

the average of $\boldsymbol{W}_{(i,j)}^{M'}$'s with $i < j$ and $(i,j) \notin \mathcal{E}$) during the training process. These gaps keep increasing until convergence, suggesting that weights between edges and non-edges are more easily separated as the learning process proceeds.

**Reachability Matrix.** In the 1-layer and 1-head Transformers, let $\boldsymbol{W}^{V'}$ be the matrix whose $i$-th row is $(\boldsymbol{e}_i^\top \boldsymbol{W}_t)\boldsymbol{W}^V \boldsymbol{W}_o + \text{FFN}\left((\boldsymbol{e}_i^\top \boldsymbol{W}_t)\boldsymbol{W}^V\right)\boldsymbol{W}_o$, where $\boldsymbol{e}_i$ is the one-hot column vector that represents the token for node $i$. Intuitively, this matrix is the remaining component in the output that contains information related to the target node. The detailed reason is also explained in Appendix D.

In Figure 6(a), we show the average weights of three different sets in the graphs: "obs" corresponds to the $\boldsymbol{W}_{(t,k)}^{V'}$'s with $t \geq k$ and $\boldsymbol{R}_{(t,k)}^{\text{obs}} = 1$; "real\obs" corresponds to the $\boldsymbol{W}_{(t,k)}^{V'}$'s with $t \geq k$, $\boldsymbol{R}_{(t,k)}^{\text{obs}} = 0$ but $\boldsymbol{R}_{(t,k)}^{\text{real}} = 1$; and "non" corresponds to the $\boldsymbol{W}_{(t,k)}^{V'}$'s with $t \geq k$ and $\boldsymbol{R}_{(t,k)}^{\text{real}} = 0$. Here we only show the results of graphs with 100 nodes and 200 nodes, since their accuracy is high enough and their attention is quite close to being concentrated on the target node. When there are more nodes, the ability to approximate the reachability matrix is not enough for us to distinguish it. From these average weights, we can see that the Transformer learns $\boldsymbol{R}^{\text{obs}}$ quite well, as for those terms in "real\obs", their weights are almost the same as those in "non". This echoes our analysis.

To further demonstrate that $\boldsymbol{R}^{\text{real}}$ is not learned as good as $\boldsymbol{R}^{\text{obs}}$, we divide the source-target node pairs $(s,t)$ in the test dataset into four categories: a) degree 0: $\boldsymbol{R}_{(t,s)}^{\text{obs}} = 1$; b) degree 1: $(s,t)$ is not of degree 0, while $s$ has at least one out-neighbor node $u$ such that $(u,t)$ is of degree 0, i.e. $\boldsymbol{R}_{(t,u)}^{\text{obs}} = 1$; c) degree 2: $(s,t)$ is not of degree 0 and 1, while $s$ has at least one out-neighbor node $u$ such that $(u,t)$ is of degree 1; d) degree 3 or more: the remaining $(s,t)$ pairs in the test dataset. Roughly speaking, in our analysis, for $(s,t)$ pairs of degree 0 or 1, we know that there is a node $u$ such that $\boldsymbol{A}_{(s,u)}^{\text{obs}} = 1$ and $\boldsymbol{R}_{(t,u)}^{\text{obs}} = 1$. Then node $u$ will have a large weight, indicating a high accuracy. As for $(s,t)$ pairs of degree 2 or more, there is no node $u$ such that both $\boldsymbol{A}_{(s,u)}^{\text{obs}} = 1$ and $\boldsymbol{R}_{(t,u)}^{\text{obs}} = 1$. In this case, the high-weight entry when predicting the next node of $s$ is either an adjacent node of $s$ or a recorded node that can reach $t$. This should reduce the accuracy.

To see this, we check the accuracy of the Transformers on the $(s,t)$ pairs of the four different categories. The results are shown in Figure 6 (b)-(d). In these figures, each row of the accuracy matrix is further divided into four sub-rows corresponding to the accuracy of degree-0 pairs, degree-1 pairs, degree-2 pairs, and degree-3 or more pairs respectively (in the graph with 100 nodes, there are no test $(s,t)$ pairs in the degree-3 or more category). From these results, we can see that the accuracy for degree-2 pairs and degree-3 or more pairs is much lower than the two other categories in most cases. It indicates that, even with more parameters and a more complex structure (e.g. a 6-layer and 6-head Transformer), the Transformer model has a fundamental difficulty in generating paths for high-degree source-target pairs, namely those pairs that can only be connected by concatenating several path segments in the training dataset. This result demonstrates the validity of our theoretical analysis, i.e., after training with gradient descent on cross-entropy loss, the Transformer can only learn observed reachability, and will miss those unobserved reachability deduced from the transitivity of the reachability relation.

In summary, our extensive empirical evaluation leads to the following conclusions about the Transformer model in achieving the path-finding task: (a) With large enough embedding size, the model can achieve high accuracy in general; (b) The model achieves its performance by concentrating attention on the target nodes as intended, and learning the information on adjacency and reachability matrices, just as what a human would do and as predicted by our theoretical analysis; and (c) The model may have limitations and fail to learn high-order reachability relations through transitivity, and thus fail to generate paths derived from high-order reachability.

# 5  Discussion and Future Work

In summary, this paper and Project ALPINE more broadly conceptualize planning as path-finding in networks, and combine theoretical analysis of the Transformer architecture and autoregressive loss with empirical validation. Our aim is to uncover the mechanisms by which intelligence may emerge from autoregressive Transformer architectures. We analytically demonstrate that Transformers possess the expressiveness required to perform path-finding tasks and that gradient descent on cross-entropy loss enables them to learn necessary—but incomplete—graph information for path-finding.

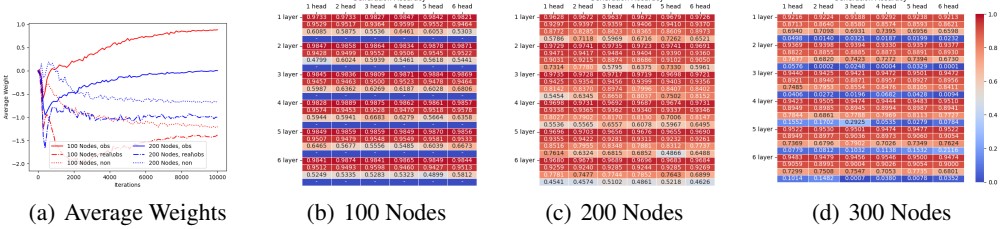

| (a) Average Weights | (b) 100 Nodes | (c) 200 Nodes | (d) 300 Nodes |

Figure 6: The average weights in $\boldsymbol{W}^{V'}$, and the accuracy for $(s,t)$'s with different degrees.

Additionally, we reveal both analytically and empirically that autoregressive training of language models has inherent limitations in the path-finding task.

**Practical Implications**: Our findings in LLMs for path planning may have practical implications for the training, testing, and enhancement of language models. In particular, the limitations we identified in current Transformer architectures for transitive reasoning suggest several directions for enhancing LLM frameworks to achieve more advanced and general planning-and-reasoning capabilities across diverse applications. For instance, in data generation for training, creating more diversified datasets that explicitly cover more reachability relationships may help the model achieve a higher accuracy. When evaluating a language model's planning capability, it may be beneficial to test for higher-order relationships not directly encoded in the training data but requiring chaining and concatenation to assess whether the model can perform transitive planning. Furthermore, by highlighting limitations in current language models, our study motivates future research into improved Transformer architectures, including incorporating transitivity directly into the model structure.

**Challenges in Reasoning about Unobserved Reachability**: Technically, the challenge in learning unobserved reachability with current Transformer architectures stems from the nature of next-token prediction loss: learning unobserved reachability incurs a higher training loss. Specifically, when predicting the next token for a given current node $i$ and target node $j$, the optimal distribution for minimizing training loss should align with the observed distribution in the training dataset, i.e., $\Pr[\text{next node} = k | \text{current node} = i \text{ and target node} = j] = \frac{N_{i,j,k}}{N_{i,j}}$ (as explained in Section 3.2). Learning unobserved reachabilities requires deviating from the distribution defined by the training data, which leads to a higher training loss. Consequently, with the current training loss and Transformer architecture, the model cannot learn unobserved reachabilities, such as transitive reachability. To enable the model to learn transitivity, we may need alternative training objectives, such as path accuracy, or structural improvements to the Transformer that allow it to 'deduce' unobserved reachabilities. Conceptually, the current training data and loss objective do not provide sufficient information to teach the model transitivity or other derived relationships. Therefore, enhancing transitivity and similar capabilities may require enriching the training data, modifying the objective function, or incorporating new components into the model architecture.

**Future Directions**: Our investigation opens several promising directions for future research: (a) Extending our study to hyper-graphs and hyper-paths, where a hyper-edge represents scenarios requiring multiple preconditions to be met simultaneously in order to carry out the next step, as often seen in task planning and mathematical proofs. (b) Addressing the limitations of Transformers in path-finding and other planning tasks by exploring richer path-finding languages, fine-tuning, or architectural improvements to LLMs. (c) Examining connections between the abstract path-finding task and concrete planning tasks (e.g., block manipulation in Blocksworld) to understand whether, and how, Transformers abstract these tasks into path-finding frameworks. (d) Investigating in-context path-finding capabilities, where training data includes different graphs with corresponding paths, to see how Transformers learn to find new paths in new graphs. (e) Exploring the integration of chain-of-thought and backtracking capabilities into Transformers for path-finding, which may offer crucial insights into enabling these features for general search and planning tasks.

In our ongoing project ALPINE, we plan to deepen our investigation into all the aforementioned fronts. We also hope that our work will inspire more researchers to study LLMs through combined theoretical and empirical analysis, with the ultimate goal of enhancing their capabilities and understanding how human-like intelligence can be achieved through statistical learning and AI mechanisms.

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

# A  Related Works

## A.1  LLMs for Planning

Several recent studies have empirically evaluated the planning abilities of large language models. For instance, CogEval has introduced a variety of planning tasks set in mazes and graphs, ultimately finding no evidence to suggest LLMs grasp the intricacies of the maps or the planning tasks themselves [16]. Similarly, another study explored the Blocksworld game, a planning challenge where humans typically achieve success rates above $70\%$, in stark contrast to GPT-3's mere $5\%$ success rate [22]. Our paper also proposes a novel approach by formulating a class of planning problems as path-finding on graphs, applying this model to the Blocksworld game and uncovering significant insights, as detailed in Appendix F.

Despite these seemingly negative evaluations, LLMs have shown remarkable aptitude in executing real-world planning tasks, creating the field of *autonomous agents* [25]. Certain applications of autonomous agents feature explicit graphs. In the tool agent HuggingGPT [18], LLMs are deployed to trigger a sequence of external APIs in response to user requests. Here, APIs are conceptualized as graph nodes, with their interrelations represented as edges, and the selection process akin to identifying a path or subgraph that aligns with user demands. This scenario is an extension of the settings discussed in this paper, where the graph is text-attributed and the objective function is evaluated through textual analysis. In addition, the application of graph search techniques has been shown to enhance the performance of tool agents significantly [13, 12]. This demonstrates that our approach of abstracting planning as path-finding in graphs is reasonable. The math agent AlphaGeometry utilizes LLMs to solve geometry problems [21]. By treating lemmas as nodes and their interdependencies as edges, the process of finding a proof of a theorem is analogous to finding a path to the theorem node in the above graph formed by possible lemma nodes and their interdependency edges. However, [21] only focuses on using LLMs to generate auxiliary constructions, and the reasoning tasks are done by a non-LLM engine. This is very different from our approach. There are no explicit graphs in other agents, such as game agents [23], embodied agents [10], and code agents [19]. The core strategy in these domains is to employ verbal reinforcement learning within LLMs. However, it is noteworthy that any dynamic programming problem characterized by deterministic state transitions can be reformulated as a shortest path problem on a graph, with states and transitions represented as nodes and edges, respectively. As a result, the area of autonomous agents is also closely related to the path-finding task investigated in this paper.

## A.2  LLMs for Graphs

GPT4Graph [9] and NLGraph [24] have developed extensive frameworks for assessing LLMs in the context of graph tasks. These frameworks encompass a broad spectrum of challenges, including classic graph problems (e.g., connectivity, cycle detection, and topological sorting), graph neural network (GNN) tasks (e.g., node and graph classification), and semantic graph question answering (e.g., knowledge graph inquiries). They also explore various input formats, such as adjacency lists, edge lists, GML, and GraphML, alongside innovative prompting techniques such as few-shot, role prompting, chain-of-thought, and algorithmic prompting (e.g., stating "we are using DFS"). These studies demonstrate that LLMs possess basic graph processing capabilities, and the choice of prompts and formats significantly influences the performance. Yet, they also reveal the models' susceptibility to spurious correlations within graphs. GPT-4, for instance, only achieves around $50\%$ accuracy on shortest path tasks, even when utilizing complex prompts. To our knowledge, our paper presents the first theoretical analysis that identifies and explains the spurious correlations learned by transformers, partially supporting some of the negative outcomes reported in these studies.

There has also been a surge in efforts aiming at bolstering LLMs' performance on graph tasks. Innovations such as GraphGPT [20] and GraphLLM [6], which incorporate an additional GNN encoder, have shown notable improvements across the aforementioned graph tasks. GraphInstruct [14] seeks to enhance LLMs' capabilities using pure LLM approaches. This involves meticulously documenting the steps of classical algorithms (e.g., BFS and DFS) and fine-tuning LLMs to learn these graph algorithms. This method of procedural supervision has extended the capacity of LLMs in graph tasks from the complexity class $TC^0$ to P/poly [7]. However, while this approach has yielded performance improvements in simpler tasks such as topological sorting and connectivity, it has proven less effective for more complex challenges, e.g., finding Hamiltonian Paths.

### A.3 Algorithm Simulation with Transformers

Recent theoretical investigations have shed light on the capability of the Transformer to simulate algorithms, a topic that has garnered considerable interest. From the view of discrete algorithms, Transformer models are likened to parallel circuits characterized by polynomial width and constant depth, which places them within the $TC^0$ complexity class (note that $TC^0 \subseteq NC^1 \subseteq P$). On the other hand, despite their impressive expressiveness, the Transformer is theoretically incapable of addressing a range of P-complete problems, including the testing of Context-Free Grammar Membership [15]. However, the advent of chain-of-thought prompting has enabled the Transformer to sequentially simulate algorithms, thereby equipping them to tackle P-complete problems in domains such as arithmetic and decision-making [7]. The exploration then extends to continuous algorithms, where it has been demonstrated that the Transformer can approximate functions such as matrix inversion, Stochastic Gradient Descent (SGD), and power iterations [8]. Our study specifically applies Transformer models to simulate path-finding algorithms, presenting evidence that their expressiveness is sufficient for such tasks (Theorem 2). Nevertheless, the usage of autoregressive loss and gradient descent introduces certain limitations, which have not been studied in existing works.

### A.4 Mechansims of LLMs

LLMs have demonstrated capabilities that exceed the theoretically predicted lower bounds of expressiveness. To demystify this paradox, numerous studies have employed experimental methodologies akin to those used in the physical and biological sciences. Their aim is to decode the mechanisms of LLMs. The foundational strategy is to generate controlled synthetic datasets to analyze how transformers (not necessarily LLMs) complete various tasks. Standard methods for this analysis include visualizing attention patterns to examine computational properties (such as locality and time invariance) and employing linear probing on the hidden states to determine the extent of learning. Given that the training data is synthetic and the ground-truth mappings are generally known, it becomes feasible to isolate the influence of various factors (e.g., prompting strategies, chain-of-thought reasoning, and data formatting). For example, a dataset designed for learning group operations, as detailed in [31], facilitates the exploration of how pretraining, data composition, and neural architecture influence reasoning tasks within LLMs. Similarly, the generation of synthetic context-free grammar (CFG) data, as described in [2], enables training GPT-2 models, uncovering their capacity to learn dynamic programming algorithms for parsing CFGs. Synthetic datasets focusing on biographical knowledge, presented in [3, 4, 5], probe into the mechanisms of knowledge storage, retrieval, manipulation, and the implications of scaling laws. Moreover, the work in [11] introduces synthetic datasets that aim to understand how smaller LLMs tackle basic arithmetic operations, e.g., addition, and examines the effects of few-shot prompting, pretraining, and model scaling [11]. Our work builds upon these investigations by conducting controlled experiments with a path-finding dataset, thereby shedding light on the complexities and challenges of planning in language models.

## B   Proof of Theorem 2

**Theorem 2.** *Given a graph $\mathcal{G}$ (with adjacency matrix $\boldsymbol{A}^{\text{true}}$ and reachability matrix $\boldsymbol{R}^{\text{true}}$), for every $\varepsilon > 0$, there exists a 1-layer, 1-head, and $O(|\mathcal{V}|)$-embedding-size Transformer model that generates a valid path for every valid source-target pair $(s, t)$ with probability at least $1 - \varepsilon$.*

*Proof.* For simplicity, we omit all layer normalizations in this construction. Suppose the input token sequence is "$s\ t\ s\ u_1\ u_2\ \ldots\ u_k$" with $k \geq 0$, where $s\ (= u_0)$ and $t$ are the tokens of the source and target nodes, respectively, and nodes $s, u_1, \cdots, u_k$ form a path that can reach node $t$ in graph $\mathcal{G}$. Our objective is to construct a 1-layer and 1-head Transformer model that generates an out-neighbor $u_{k+1}$ of $u_k$ such that there exists at least one path from $u_{k+1}$ to $t$ in $\mathcal{G}$.

In essence, we utilize the attention layer to attend the output *solely* to the target node $t$. This approach allows the distribution of next token $u_{k+1}$ to become a function of both the current node $u_k$ and the target node $t$ (as formulated in Section 2). Then, by integrating the adjacency matrix $\boldsymbol{A}^{\text{true}}$ into the FFN layer and the reachability matrix $\boldsymbol{R}^{\text{true}}$ into the matrix $\boldsymbol{W}^V$ in the attention layer, we extract row vectors $\boldsymbol{R}^{\text{true}}_{(t,:)}$ and $\boldsymbol{A}^{\text{true}}_{(u_k,:)}$ from $\boldsymbol{R}^{\text{true}}$ and $\boldsymbol{A}^{\text{true}}$, respectively, corresponding to the target node $t$ and current node $u_k$. By selecting proper coefficients, we can let the output be the sum of $\boldsymbol{R}^{\text{true}}_{(t,:)}$ and

$\boldsymbol{A}^{\text{true}}_{(u_k,:)}$. Following the softmax layer, the non-negligible entries in the final vector correspond to the feasible next nodes. With this encoding, the Transformer serves as a simulator of Algorithm 1 with input $\boldsymbol{A} = \boldsymbol{A}^{\text{true}}$ and $\boldsymbol{R} = \boldsymbol{R}^{\text{true}}$.

Following our notation in Section 2.1, we adopt $d = |\mathcal{V}| + 2$, $M = |\mathcal{V}| + 1$ and $N = k + 3$. In the Transformer, there are $M = |\mathcal{V}| + 1$ tokens representing the $|\mathcal{V}|$ nodes and the end-of-line "\n". Hence, the input tokens can be represented by the one-hot embedding matrix $\boldsymbol{U} \in \mathbb{R}^{N \times M}$. We let $\boldsymbol{W}_t = (\boldsymbol{I}_{M \times M} \mid \boldsymbol{0}_{M \times 1}) \in \mathbb{R}^{M \times d}$ and $\boldsymbol{W}_p = (\boldsymbol{0}_{(k+3) \times (|\mathcal{V}|+1)} \mid c_0 \cdot \boldsymbol{e}_2) \in \mathbb{R}^{N \times d}$, here $\boldsymbol{e}_2$ represents the second unit column vector of dimension $k + 3$, $(A \mid B)$ is the notation for matrix concatenation by column, and $c_0$ is a positive parameter to be decided. According to the definition of the Transformer, we now have a matrix $\boldsymbol{H}_0$ such that the first $|\mathcal{V}| + 1$ columns are the tokens of nodes in the sequence and the last column indicates the positions of the target node $t$. More specifically, we have

$$
\boldsymbol{H}_0 = \begin{pmatrix} \boldsymbol{e}_s^\top & 0 \\ \boldsymbol{e}_t^\top & c_0 \\ \boldsymbol{e}_s^\top & 0 \\ \boldsymbol{e}_{u_1}^\top & 0 \\ \cdots & \cdots \\ \boldsymbol{e}_{u_k}^\top & 0 \end{pmatrix} \in \mathbb{R}^{N \times d},
$$

here $\boldsymbol{e}_u$ represents the one-hot token vector for node $u$ (with dimension $M = |\mathcal{V}| + 1$).

Then we construct the attention layer of the Transformer. We only have one head and let $\boldsymbol{W}^K = (\boldsymbol{0}_{(|\mathcal{V}|+2) \times (|\mathcal{V}|+1)} \mid \boldsymbol{1}_{(|\mathcal{V}|+2) \times 1})^\top \in \mathbb{R}^{d \times d}$ and $\boldsymbol{W}^Q = \sqrt{d} \cdot \boldsymbol{I}_{d \times d}$. Then we can compute $\boldsymbol{H}_0 \boldsymbol{W}^K = (\boldsymbol{0}_{(|\mathcal{V}|+2) \times 1} \mid c_0 \cdot \boldsymbol{1}_{(|\mathcal{V}|+2) \times 1} \mid \boldsymbol{0}_{(|\mathcal{V}|+2) \times (k+1)})^\top$, i.e., second rows are all $c_0$'s and other rows are all 0's, and $\boldsymbol{H}_0 \boldsymbol{W}^Q = \sqrt{d} \cdot \boldsymbol{H}_0$.

Therefore,

$$
\frac{(\boldsymbol{H}_0 \boldsymbol{W}^Q)(\boldsymbol{H}_0 \boldsymbol{W}^K)^\top}{\sqrt{d}} = \begin{pmatrix} 0 & c_0 & \boldsymbol{0}_{1 \times (k+1)} \\ 0 & c_0^2 + c_0 & \boldsymbol{0}_{1 \times (k+1)} \\ \boldsymbol{0}_{(k+1) \times 1} & c_0 \cdot \boldsymbol{1}_{(k+1) \times 1} & \boldsymbol{0}_{(k+1) \times (k+1)} \end{pmatrix} \in \mathbb{R}^{N \times N}.
$$

And we can compute the first part of the attention layer as

$$
\mathbf{softmax}\left( \frac{(\boldsymbol{H}_0 \boldsymbol{W}^Q)(\boldsymbol{H}_0 \boldsymbol{W}^K)^\top}{\sqrt{d}} \right) = \begin{pmatrix} \frac{1}{k+2+e^{c_0}} & \frac{e^{c_0}}{k+2+e^{c_0}} & \frac{1}{k+2+e^{c_0}} \cdot \boldsymbol{1}_{1 \times (k+1)} \\ \frac{1}{k+2+e^{c_0^2+c_0}} & \frac{e^{c_0^2+c_0}}{k+2+e^{c_0^2+c_0}} & \frac{1}{k+2+e^{c_0^2+c_0}} \cdot \boldsymbol{1}_{1 \times (k+1)} \\ \frac{1}{k+2+e^{c_0}} \cdot \boldsymbol{1}_{(k+1) \times 1} & \frac{e^{c_0}}{k+2+e^{c_0}} \cdot \boldsymbol{1}_{(k+1) \times 1} & \frac{1}{k+2+e^{c_0}} \cdot \boldsymbol{1}_{(k+1) \times (k+1)} \end{pmatrix} \in \mathbb{R}^{N \times N}.
$$

By setting $c_0 \to +\infty$, we obtain:

$$
\mathbf{softmax}\left( \frac{(\boldsymbol{H}_0 \boldsymbol{W}^Q)(\boldsymbol{H}_0 \boldsymbol{W}^K)^\top}{\sqrt{d}} \right) \to \begin{pmatrix} 0 & 1 & \boldsymbol{0}_{1 \times (k+1)} \\ \cdots & \cdots & \cdots \\ 0 & 1 & \boldsymbol{0}_{1 \times (k+1)} \end{pmatrix}.
$$

Furthermore, we set $\boldsymbol{W}^V = \begin{pmatrix} c_1 \cdot \boldsymbol{R}^{\text{true}} & \boldsymbol{0}_{|\mathcal{V}| \times 2} \\ \boldsymbol{0}_{2 \times |\mathcal{V}|} & \boldsymbol{0}_{2 \times 2} \end{pmatrix}$, where $c_1 > 0$ is also a parameter to be decided later. Then after the attention layer, we have a matrix as

$$
\lim_{c_0 \to +\infty} \mathrm{MHA}(\boldsymbol{H}_0) = c_1 \cdot \begin{pmatrix} \boldsymbol{R}^{\text{true}}_{(t,:)} & 0 & 0 \\ \cdots & \cdots & \cdots \\ \boldsymbol{R}^{\text{true}}_{(t,:)} & 0 & 0 \end{pmatrix} \in \mathbb{R}^{N \times d}.
$$

Now we construct the feed-forward layer, which is a two-layer MLP.

For the first layer, the weight matrix $\boldsymbol{W}_1$ is set to be,

$$
\boldsymbol{W}_1 = \begin{pmatrix} \boldsymbol{I}_{(|\mathcal{V}|+2) \times (|\mathcal{V}|+2)} & \boldsymbol{0}_{(|\mathcal{V}|+2) \times 3(|\mathcal{V}|+2)} \end{pmatrix} \in \mathbb{R}^{d \times 4d}.
$$

and the bias $\boldsymbol{b}_1 = -c_1 \cdot \boldsymbol{1}_{4d \times 1}$, which implies that $\boldsymbol{1}_{N \times 1} \boldsymbol{b}_1^\top = -c_1 \cdot \boldsymbol{1}_{N \times 4d}$. When $c_0$ is large enough, the $(k+3)^{th}$ row of the matrix $\max\left( \boldsymbol{0}, (\mathrm{MHA}(\boldsymbol{H}_0) + \boldsymbol{H}_0) \boldsymbol{W}_1 + \boldsymbol{1}_{N \times 1} \boldsymbol{b}_1^\top \right)$

is $\max\left(\mathbf{0}, c_1 \cdot \left(\mathbf{R}_{(t,:)} \mid \mathbf{0}_{1\times(3|\mathcal{V}|+8)}\right) + \left(\mathbf{e}_{u_k}^\top \mid \mathbf{0}_{1\times(3|\mathcal{V}|+7)}\right) - c_1 \cdot \mathbf{1}_{1\times4(|\mathcal{V}|+2)}\right)$. Since $u_k$ can reach $t$, in $c_1 \cdot \left(\mathbf{R}_{(t,:)} \mid \mathbf{0}_{1\times(3|\mathcal{V}|+8)}\right) + \left(\mathbf{e}_{u_k}^\top \mid \mathbf{0}_{1\times(3|\mathcal{V}|+7)}\right)$, only the entry for node $u_k$ is $c_1 + 1$ while all other entries are $0$ or $c_1$. Therefore, the $(k+3)^{th}$ row of the matrix $\max\left(\mathbf{0}, (\mathrm{MHA}(\mathbf{H}_0) + \mathbf{H}_0)\mathbf{W}_1 + \mathbf{1}_{N\times1}\mathbf{b}_1^\top\right)$ can be arbitrarily close to $\left(\mathbf{e}_{u_k}^\top \mid \mathbf{0}_{1\times(3|\mathcal{V}|+7)}\right)$. Here $\mathbf{e}_u$ represents the one-hot token vector for node $u$ (with dimension $M = |\mathcal{V}| + 1$).

For the second layer, we set

$$\mathbf{W}_2 = \begin{pmatrix} c_2 \cdot \mathbf{A} & \mathbf{0}_{|\mathcal{V}|\times2} \\ \mathbf{0}_{(3|\mathcal{V}|+8)\times|\mathcal{V}|} & \mathbf{0}_{(3|\mathcal{V}|+8)\times2} \end{pmatrix} \in \mathbb{R}^{4d\times d},$$

where $c_2$ is a positive parameter to be decided, and $\mathbf{b}_2 = \mathbf{0}$. By this way, we have

$$\lim_{c_0\to\infty} (\mathrm{FFN}(\mathrm{MHA}(\mathbf{H}_0) + \mathbf{H}_0))_{(k+3,:)} \to \left(c_2 \cdot \mathbf{A}_{(u_k,:)} \mid \mathbf{0}_{1\times2}\right) \in \mathbb{R}^{|\mathcal{V}|+2}.$$

Therefore,

$$\lim_{c_0\to\infty} (\mathbf{H}_1)_{(k+3,:)} \to \left(c_1 \cdot \mathbf{R}_{(t,:)} + c_2 \cdot \mathbf{A}_{(u_j,:)} \mid \mathbf{0}_{1\times2}\right) + \left(\mathbf{e}_{u_k} \mid 0\right) \in \mathbb{R}^{|\mathcal{V}|+2},$$

where $\mathbf{e}_u$ represents the one-hot token vector for node $u$ (with dimension $M = |\mathcal{V}| + 1$).

Then we fix $c_1 = c_2$ and let them be large enough. In this case, the dominant entries in $(\mathbf{H}_1)_{(k+3,:)}$ represent the nodes that are both the out-neighbor of $u_j$ and reachable to $t$, since those entries will have the value of $2c_1$ while other entries are at most $c_1 + 1$. This means that $(\mathbf{H}_1)_{(k+3,:)}$ can correctly indicates the next node $u_{k+1}$. Specifically, let $\mathbf{W}_o = \left(\mathbf{I}_{(|\mathcal{V}|+1)\times(|\mathcal{V}|+1)} \mid \mathbf{0}_{(|\mathcal{V}|+1)\times1}\right)^\top \in \mathbb{R}^{d\times M}$. Then the final output approaches the following vector

$$\lim_{c_0,c_1=c_2\to\infty} \hat{\mathbf{u}}_{k+1} = \lim_{c_0,c_1=c_2\to\infty} \mathrm{softmax}((\mathbf{H}_1)_{(k+3,:)}\mathbf{W}_o)$$

$$= \frac{1}{C} \cdot \left(\mathbb{I}[\mathbf{A}_{(u_k,1)} = 1 \wedge \mathbf{R}_{(t,1)} = 1], \cdots, \mathbb{I}[\mathbf{A}_{(u_k,|\mathcal{V}|)} = 1 \wedge \mathbf{R}_{(t,|\mathcal{V}|)} = 1], 0\right),$$

where $C$ is the number of nodes that are both the out-neighbor of $u_k$ and reachable to $t$. Thus, this encoding guarantees that for any $\varepsilon > 0$ and $Q > 0$, we can always construct a 1-layer, 1-head, and $(|\mathcal{V}| + 2)$-embedding-size Transformer that provides the correct next token with probability at least $1 - \frac{\varepsilon}{2Q}$ by selecting large enough parameters $c_0, c_1, c_2$.

Then we prove that there exists a $Q$ such that this Transformer can output a correct path for every valid source and target node pair with probability at least $1 - \varepsilon$. Suppose we can output all the nodes that are both the out-neighbor of the current node and reachable to the target node with the same probability in each round without any error. Then, whatever the current node is, there is a probability of at least $\frac{1}{|\mathcal{V}|^{|\mathcal{V}|}}$ that the target node is reached within the next $|\mathcal{V}|$ generated nodes. Therefore, the target node is reached in $c_3 \cdot |\mathcal{V}|$ steps with probability at least $1 - \left(1 - \frac{1}{|\mathcal{V}|^{|\mathcal{V}|}}\right)^{c_3}$, where $c_3 \in \mathbb{N}$ is a positive integer. We let $c_3 = \log_{1-\frac{1}{|\mathcal{V}|^{|\mathcal{V}|}}} \frac{\varepsilon}{2}$ and $Q = c_3 \cdot |\mathcal{V}|$. Then, according to the Union bound, the Transformer can output a correct path in $Q$ steps with an error rate of at most $\frac{\varepsilon}{2Q} \cdot Q + \frac{\varepsilon}{2} = \varepsilon$.

Finally, there are two different rules (other than output a correct next node): i) when the input sequence is only "$s$ $t$", the prediction of the next token should be the source node $s$; ii) when the input sequence is "$s$ $t$ $s$ $a$ $b$ $c$ $t$", the prediction of the next token should be \n. Case i) can be solved using the Transformer architecture utilizing the position information and attention to the first position; and case ii) can be solved by using the Transformer architecture utilizing the position information and attention to the second position. To maintain focus on the main construction corresponding to Algorithm 1, we omit the detailed construction for these two boundary cases. $\square$

## C   Proof of Theorem 3

**Theorem 3.** *Under the cross-entropy loss $\ell(\mathcal{D})$, for all $(i, k)$ pairs, i) if $\sum_j N_{i,j} = 0$, then $\frac{\partial \ell(\mathcal{D})}{\partial \mathbf{W}_{(i,k)}^M}$ is always 0; ii) if $\sum_j N_{i,j} > 0$ but $\sum_j N_{i,j,k} = 0$, then $\frac{\partial \ell(\mathcal{D})}{\partial \mathbf{W}_{(i,k)}^M}$ is always positive; iii) if $\sum_j N_{i,j,k} > 0$, then $\frac{\partial \ell(\mathcal{D})}{\partial \mathbf{W}_{(i,k)}^M}$ is negative when $\mathbf{W}_{(i,k)}^M \to -\infty$. Similarly, for all $(j, k)$ pairs, i) if $\sum_i N_{i,j} = 0$, then*

$\frac{\partial \ell(\mathcal{D})}{\partial \boldsymbol{W}_{(j,k)}^{V}}$ is always 0; ii) if $\sum_i N_{i,j} > 0$ but $\sum_i N_{i,j,k} = 0$, then $\frac{\partial \ell(\mathcal{D})}{\partial \boldsymbol{W}_{(j,k)}^{V}}$ is always positive; iii) if $\sum_i N_{i,j,k} > 0$, then $\frac{\partial \ell(\mathcal{D})}{\partial \boldsymbol{W}_{(j,k)}^{V}}$ is negative when $\boldsymbol{W}_{(j,k)}^{V} \to -\infty$.

*Proof.* We only prove the first part of this theorem, since the proof of the second part is almost identical.

By the definition of the cross-entropy loss in Eq.(4), and the prediction weight vector in Eq.(5) for our simplified model, the total cross-entropy loss of the model (with matrices $\boldsymbol{W}^M$, $\boldsymbol{W}^V$) is

$$
\begin{aligned}
\ell(\mathcal{D}) &= -\sum_{\boldsymbol{u}\in\mathcal{D}}\sum_{n\geq 3}\sum_{k} \boldsymbol{U}_{(n+1,k)} \log \hat{\boldsymbol{u}}_{(n+1),k} \\
&= -\sum_{\boldsymbol{u}\in\mathcal{D}}\sum_{n\geq 3}\sum_{k} \boldsymbol{U}_{(n+1,k)} \log \frac{\exp(\boldsymbol{W}_{(u_n,k)}^M + \boldsymbol{W}_{(u_2,k)}^V)}{\sum_{\ell}\exp(\boldsymbol{W}_{(u_n,\ell)}^M + \boldsymbol{W}_{(u_2,\ell)}^V)} \\
&= -\sum_{\boldsymbol{u}\in\mathcal{D}}\sum_{n\geq 3}\sum_{k} \boldsymbol{U}_{(n+1,k)} \sum_{i,j}\mathbb{I}[u_n=i,u_2=j] \log \frac{\exp(\boldsymbol{W}_{(i,k)}^M + \boldsymbol{W}_{(j,k)}^V)}{\sum_{\ell}\exp(\boldsymbol{W}_{(i,\ell)}^M + \boldsymbol{W}_{(j,\ell)}^V)} \\
&= -\sum_{i,j,k} N_{i,j,k} \log \frac{\exp(\boldsymbol{W}_{(i,k)}^M + \boldsymbol{W}_{(j,k)}^V)}{\sum_{\ell}\exp(\boldsymbol{W}_{(i,\ell)}^M + \boldsymbol{W}_{(j,\ell)}^V)} \\
&= -\sum_{i,j,k} N_{i,j,k}(\boldsymbol{W}_{(i,k)}^M + \boldsymbol{W}_{(j,k)}^V) + \sum_{i,j,k} N_{i,j,k} \log\left(\sum_{\ell}\exp(\boldsymbol{W}_{(i,\ell)}^M + \boldsymbol{W}_{(j,\ell)}^V)\right) \\
&= -\sum_{i,j,k} N_{i,j,k}(\boldsymbol{W}_{(i,k)}^M + \boldsymbol{W}_{(j,k)}^V) + \sum_{i,j} N_{i,j} \log\left(\sum_{\ell}\exp(\boldsymbol{W}_{(i,\ell)}^M + \boldsymbol{W}_{(j,\ell)}^V)\right).
\end{aligned}
$$

Then we have that

$$
\frac{\partial \ell(\mathcal{D})}{\partial \boldsymbol{W}_{(i,k)}^M} = -\sum_{j} N_{i,j,k} + \sum_{j} N_{i,j} \frac{\exp(\boldsymbol{W}_{(i,k)}^M + \boldsymbol{W}_{(j,k)}^V)}{\sum_{\ell}\exp(\boldsymbol{W}_{(i,\ell)}^M + \boldsymbol{W}_{(j,\ell)}^V)}. \tag{7}
$$

In case i), $\sum_j N_{i,j} = 0$ implies that $\sum_j N_{i,j,k} = 0$. Hence $\frac{\partial \ell(\mathcal{D})}{\partial \boldsymbol{W}_{(i,k)}^M}$ is always 0.

In case ii), $\sum_j N_{i,j} > 0$ implies that the second term in Eq. (7) is positive, while $\sum_j N_{i,j,k} = 0$ implies that the first term in Eq. (7) is 0. Hence $\frac{\partial \ell(\mathcal{D})}{\partial \boldsymbol{W}_{(i,k)}^M}$ is always positive.

In case iii), when $\sum_j N_{i,j} > 0$ and $\boldsymbol{W}_{(i,k)}^M$ converges to $-\infty$, then the second term in Eq. (7) converges to 0, and it is smaller than $\sum_j N_{i,j,k} > 0$. Hence, $\frac{\partial \ell(\mathcal{D})}{\partial \boldsymbol{W}_{(i,k)}^M}$ is negative when $\boldsymbol{W}_{(i,k)}^M$ converges to $-\infty$. $\square$

# D  The Reason of Choosing $W^{M'}$ and $W^{V'}$

Note that in the Transformer layer, the output can be written as[3]

$$
\text{FFN}\left(\text{softmax}\left(\frac{\boldsymbol{Q}\boldsymbol{K}^\top}{\sqrt{d_k}}\right)\boldsymbol{V} + \boldsymbol{X}\right) + \text{softmax}\left(\frac{\boldsymbol{Q}\boldsymbol{K}^\top}{\sqrt{d_k}}\right)\boldsymbol{V} + \boldsymbol{X}.
$$

Also noting that we have verified that the attention is concentrated at the second token, then we let $\boldsymbol{X}_2 = \boldsymbol{U}_{(2,:)}\boldsymbol{W}_t$, representing the token embedding of the target node, and $\boldsymbol{X}_n = \boldsymbol{U}_{(n,:)}\boldsymbol{W}_t$, representing the token embedding of the current node.

---

[3]For simplicity, though the layer normalizations $\text{LN}_1, \text{LN}_2$ and $\text{LN}_t$ are used in our experiments, we omit them in the equations in this section.

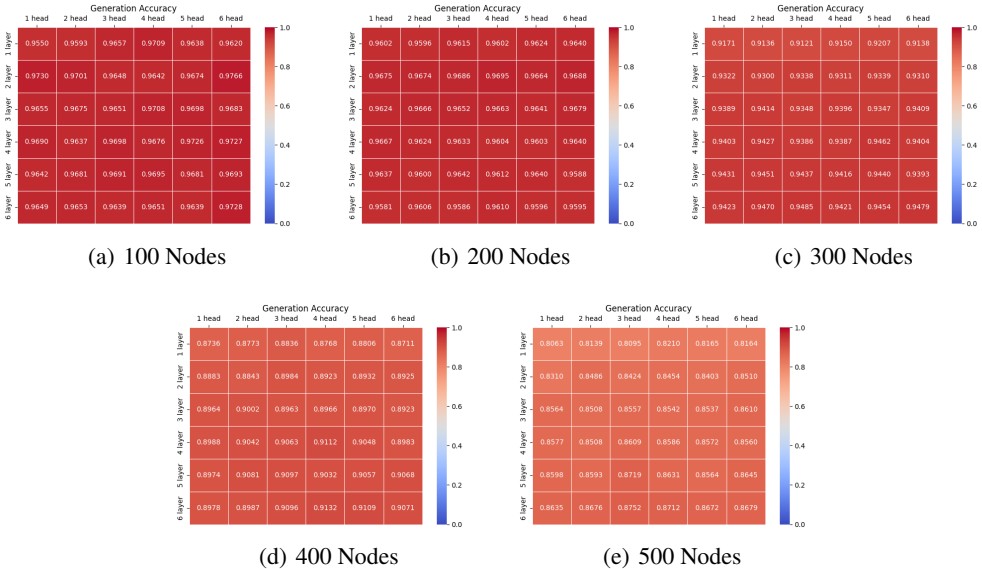

Figure 7: Accuracy on the test datasets with embedding size $d = 120$.

Then we know that

$$\hat{\boldsymbol{u}}_{(n+1)} \approx \left( \text{FFN}\left(\boldsymbol{X}_2\boldsymbol{W}^V + \boldsymbol{X}_n\right) + \boldsymbol{X}_2\boldsymbol{W}^V + \boldsymbol{X}_n \right)\boldsymbol{W}_o.$$

Table 1: Cosine Similarity of $\text{FFN}\left(\boldsymbol{X}_2\boldsymbol{W}^V + \boldsymbol{X}_n\right)\boldsymbol{W}_o$ and $\text{FFN}\left(\boldsymbol{X}_2\boldsymbol{W}^V\right)\boldsymbol{W}_o + \text{FFN}\left(\boldsymbol{X}_n\right)\boldsymbol{W}_o$

| Graph | 100 Nodes | 200 Nodes | 300 Nodes | 400 Nodes | 500 Nodes |
|---|---|---|---|---|---|
| **Average Cosine Similarity** | 0.926 | 0.924 | 0.901 | 0.870 | 0.889 |

It is straightforward that $\boldsymbol{X}_n\boldsymbol{W}_o$ contains the information of the current node, and $\boldsymbol{X}_2\boldsymbol{W}^V\boldsymbol{W}_o$ contains the information of the target node. As for $\text{FFN}\left(\boldsymbol{X}_2\boldsymbol{W}^V + \boldsymbol{X}_n\right)\boldsymbol{W}_o$, we choose to use its linear approximation as $\text{FFN}\left(\boldsymbol{X}_2\boldsymbol{W}^V + \boldsymbol{X}_n\right)\boldsymbol{W}_o \approx \text{FFN}\left(\boldsymbol{X}_2\boldsymbol{W}^V\right)\boldsymbol{W}_o + \text{FFN}\left(\boldsymbol{X}_n\right)\boldsymbol{W}_o$. As shown in Table 1 (which takes average over all possible $\boldsymbol{X}_2$'s and $\boldsymbol{X}_n$'s), this is a good approximation. Then we can treat $\text{FFN}\left(\boldsymbol{X}_2\boldsymbol{W}^V\right)\boldsymbol{W}_o$ as the information of the target node, and $\text{FFN}\left(\boldsymbol{X}_n\right)\boldsymbol{W}_o$ as the information of the current node.

Because of this, we let $\boldsymbol{W}^{M'}$ be the matrix whose $i$-th row is $\text{FFN}\left(\boldsymbol{e}_i^\top\boldsymbol{W}_t\right)\boldsymbol{W}_o + (\boldsymbol{e}_i^\top\boldsymbol{W}_t)\boldsymbol{W}_o$, where $\boldsymbol{e}_i$ represents the one-hot column vector for node $i$ (with dimension $M = |\mathcal{V}| + 1$). Note that in the simplified Transformer model of Theorem 3, there is no $\boldsymbol{X}_n\boldsymbol{W}_o$ term, and $\boldsymbol{W}^{M'}$ is the same as matrix $\boldsymbol{W}^M$. Similarly, we let $\boldsymbol{W}^{V'}$ be the matrix whose $i^{th}$ row is $(\boldsymbol{e}_i^\top\boldsymbol{W}_t)\boldsymbol{W}^V\boldsymbol{W}_o + \text{FFN}\left((\boldsymbol{e}_i^\top\boldsymbol{W}_t)\boldsymbol{W}^V\right)\boldsymbol{W}_o$, where $\boldsymbol{e}_i$ represents the one-hot column vector for node $i$ (with dimension $M = |\mathcal{V}| + 1$). Note that in the simplified Transformer model of Theorem 3, there is no $\text{FFN}\left(\boldsymbol{X}_2\boldsymbol{W}^V\right)\boldsymbol{W}_o$ term, and $\boldsymbol{W}^{V'}$ is the same as matrix $\boldsymbol{W}^V$.

## E    Complete Experimental Results on the Synthetic Datasets

In this section, we state the complete experimental results on the synthetic datasets. All these results are conducted on a single A100 GPU.

- The accuracy results on all these tests are presented in Figure 7.
- The attention results on all the 1-layer and 1-head Transformers are presented in Figure 8.
- The results of $\boldsymbol{W}^{M'}$'s are shown in Figure 9.
- The accuracy of the Transformers on the $(s, t)$ pairs of the four different categories are shown in Figure 10.

As we can see, all these results are consistent with our conclusions.

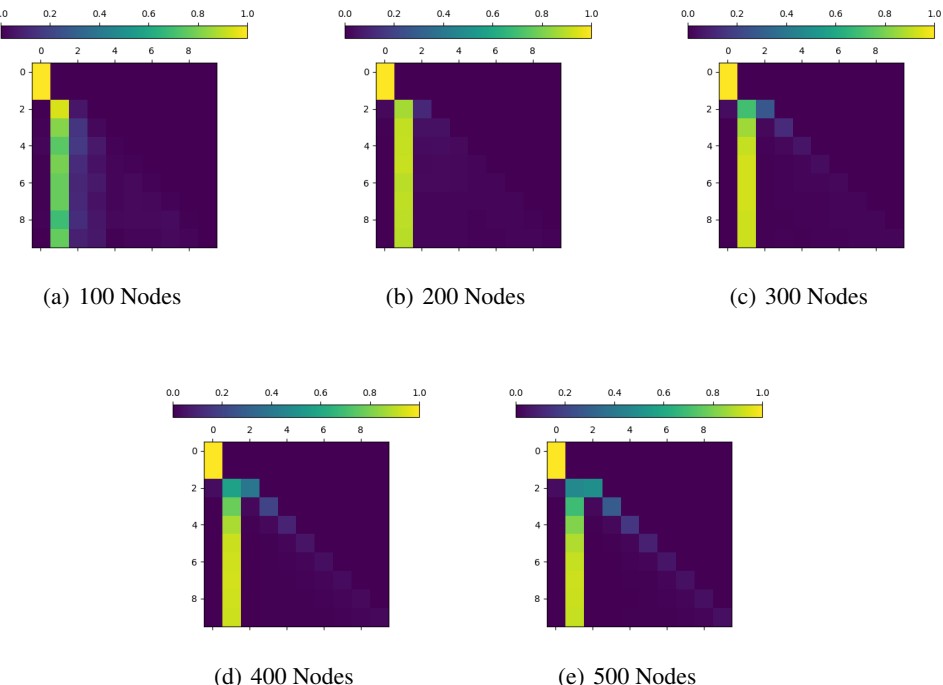

Figure 8: The average attention in 1-layer and 1-head Transformers.

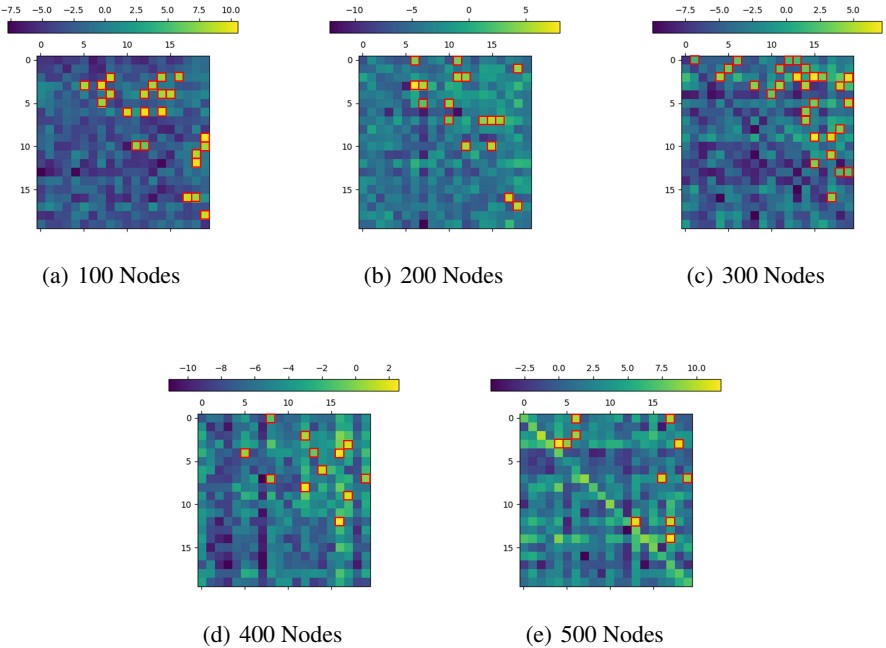

Figure 9: The first 20 rows and columns of $\boldsymbol{W}^{M'}$ matrix in the 1-layer and 1-head Transformers (the red boxes correspond to 1's in the adjacency matrix $\boldsymbol{A}$).

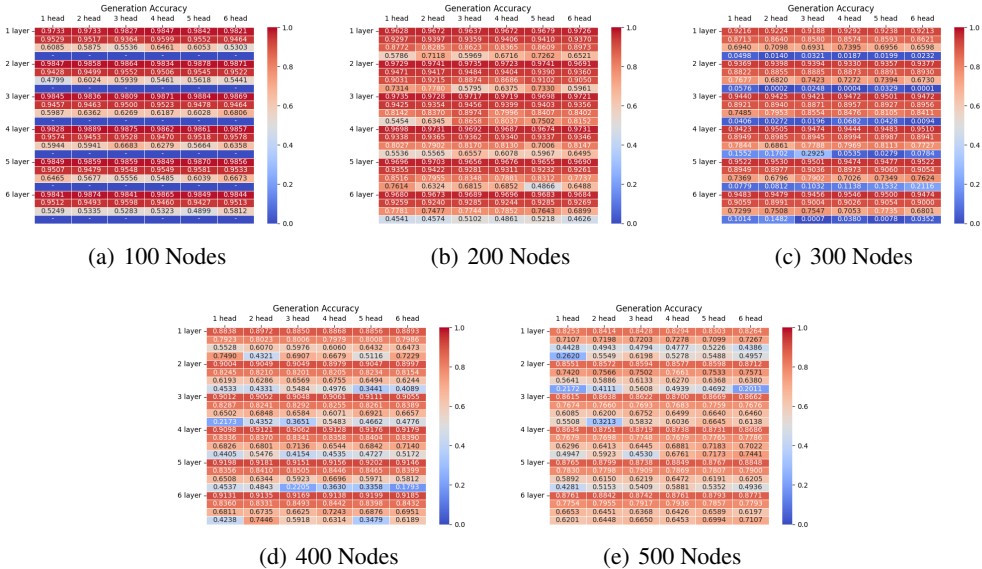

Figure 10: The accuracy for $(s,t)$'s with different degrees.

## F  Path-planning in Blocksworld

To further validate the theoretical results in Section 3 and the practicability of the proposed path-finding task, we consider Blocksworld benchmark [22]. Blocksworld is a scenario consisting of a set of blocks identified by different colors. The blocks are either placed on table or on top of another block and the task is to plan a block manipulation from the source state to the target state.

We formulate Blocksworld as a path-finding task. Here we construct a graph $G_{BW}$ for the case with 4 blocks, where each node represents a state of the blocks. For example, node 0 refers to the state that "the red block is on top of the blue block, the blue block is on top of the orange block, the orange block is on top of the yellow block, and the yellow block is on the table". $G_{BW}$ is a directed graph with 73 nodes, and the adjacency matrix of $G_{BW}$ is presented in Figure 11(a).

In the original Blocksworld task, the answer is a sequence of actions, which is equivalent to the notion of edges in $G_{BW}$. We reformulated it to let the model output a path from the given source state to the given target state, only consisting of the nodes. This can be seen as a simplified version and a pure planning task. We randomly select 80% of all node pairs for training and the rest 20% for testing, and generate 50000 training sequences in the same format as introduced in Section 2. We mainly use Transformers with 1 layer and 1 head for the convenience of visualization.

### F.1  Results

We first present the accuracy results during training when using different embedding sizes $d \in \{30, 60, 120\}$. As shown in Figure 11(d), although a smaller embedding size results in a longer time to converge, all models reach an accuracy near 100% at the end of the training.

Then, we use the same method introduced in Section 4 to visualize the attention map and the $\boldsymbol{W}^{M'}$ matrix for the model with $d = 120$ after the entire iterations. In Figure 11(c), we can see that when predicting the tokens on the path, almost all the attention weights are on the second token which represents for the target node, demonstrating the capability of the model to learn a correct attention. For adjacency matrix, we find that the $\boldsymbol{W}^{M'}$ matrix in Figure 11(b) almost perfectly aligns to the real adjacency matrix of $G_{BW}$. And the weight gap (average edge weight minus average non-edge weight) for all models keeps increasing in the training process until convergence, as shown in Figure 11(e).

In addition, we present the results related to observed reachability matrix in Figure 12. Figure 12(a) shows the observed reachability in the training dataset. Although $G_{BW}$ is fully-connected, some reachability are not observed since we request that all training data has no cycle. Specifically, in this

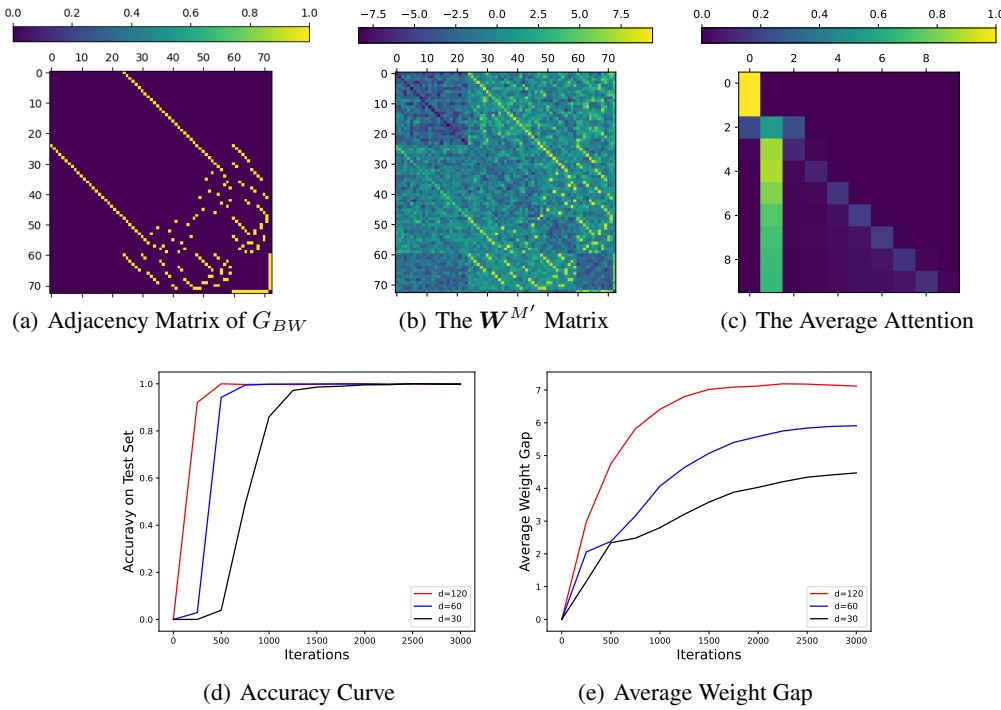

(a) Adjacency Matrix of $G_{BW}$  (b) The $\boldsymbol{W}^{M'}$ Matrix  (c) The Average Attention

(d) Accuracy Curve  (e) Average Weight Gap

Figure 11: Accuracy, attention, and adjacency matrix results for the experiment on Blocksworld benchmark.

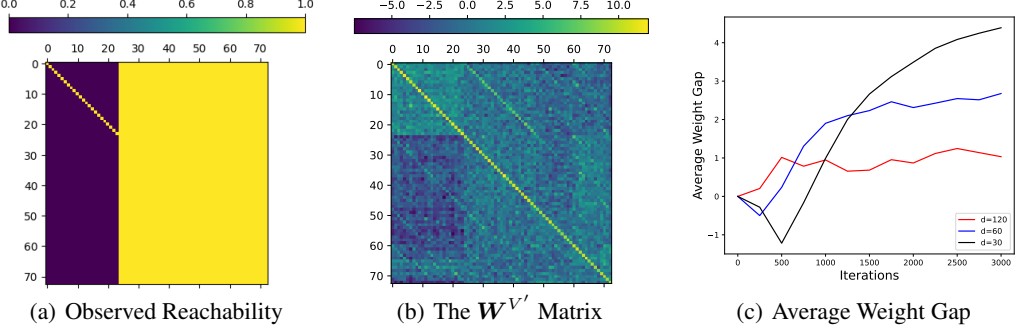

(a) Observed Reachability  (b) The $\boldsymbol{W}^{V'}$ Matrix  (c) Average Weight Gap

Figure 12: Experiment for reachability on Blocksworld benchmark.

case, each of the first 24 nodes is not observed reachable to any nodes other than itself. To validate whether the Transformer has captured this information, we construct $\boldsymbol{W}^{V'}$ matrix through the same method presented in Section 4. As shown in Figure 12(b), the first 24 columns of the $\boldsymbol{W}^{V'}$ matrix are noticeably darker, which aligns with the observed reachability matrix in Figure 12(a). Furthermore, we plot the gap between the average weight of $\boldsymbol{W}^{V'}$ on observed reachability and the average weight of $\boldsymbol{W}^{V'}$ on non-observed reachability in Figure 12(c), and find that this gap keeps increasing for all models. Since there does not exist any test pairs with degree 2 or more (as defined in Section 4), we do not compare the accuracy between different degrees in Blocksworld.

In summary, our experimental results on the Blocksworld benchmark confirm our theoretical analyses (Theorem 3) and empirical results on the synthetic data, and they at least partially explain the planning capability of the Transformer on the Blocksworld scenario.

