# OpenReview forum: "ALPINE: Unveiling The Planning Capability of Autoregressive Learning in Language Models"
_NeurIPS.cc/2024/Conference — NeurIPS 2024 poster_

### Official Review · Reviewer_wPpp · 2024-07-12

**Soundness:** 3
**Presentation:** 3
**Contribution:** 2
**Rating:** 6
**Confidence:** 4

**Summary:**

This paper addresses the question of whether or not a transformer-based language model can learn to plan based on next-token prediction. The authors analyse the ability of a transformer-based language model to learn in an abstracted path planning domain, and show that the model can some of the underlying structure of planning problems but has real difficulty in learning to generalise from the training data to be able to solve novel planning problems using relations such as transitivity.

**Strengths:**

The overall idea of this paper is pretty strong.
- The specific problem the authors address is important and relevant, especially determining if a transformer can learn to solve novel planning problems by inferring unobserved relations.
- The formulation of the analysis is reasonably strong. The combination of Theorem 2 and Theorem 3 are a statement about the expressive capacity of the model, but the limitations on the learning process.
- The experimental results do a reasonable job of supporting the theoretical analysis and claims of the paper.

**Weaknesses:**

There are unfortunately two fairly substantial weaknesses in this paper.

- Firstly, while the paper demonstrates that the specific network architecture chosen cannot deduce the existence of reachability relations that are not observed in the training data, the paper does not adequately describe why there might be any reason to think this. The result that reachability cannot be inferred without observing it is not really surprising, and the brief motivation in the third paragraph of the introduction (describing learned planning) is not really adequate. At the same time, it is a little surprising that the learner does not generalise at all to *similar* reachable concepts.

This weakness could perhaps be addressed in a revised version of the paper through a clearer introduction, and the introduction itself is a little hard to follow. I did not entirely understand where the paper was headed until I got to the end of the paper and read "the Transformer can only learn observed reachability, and will miss those unobserved reachability deduced from the transitivity of the reachability relation." (To be fair, this idea is also present in the abstract, but not strongly present in the introduction.) A clearer motivation for the investigation and why it is reasonable to be unsure about what the Transformer is learning would be extremely helpful.

- The second weakness is that it is not clear the extent to which the lack of inference of reachability is a problem with the specific network architecture. I was surprised that the feed forward layer is a multi-layer perceptron, rather than a graph neural network, as described by Khan et al (2020). Gama et al (2019) showed that GNNs are invariant to graph permutations which make them particularly useful in certain kinds of planning problems. The real problem is that the theoretical result is a positive result about what *is* learnable, and is supported by the experiments, but the primary conclusion of the paper is a negative result. The paper does not have a corresponding theoretical result to justify the negative result, and it is hard not to wonder if the experimental results are an accident of the specific network and training process.

The fact that the "Transformer model has a fundamental difficulty in generating paths for high-degree source-target pairs" may also be an accident of the network architecture, although a GNN formulation would (most likely) need to know ahead of time the degree of the node.

- The experimental results are reasonable, but the specific planning domains are somewhat ad hoc and not very general. I am happy to see assessment across a range of domain sizes, but the domains are still quite limited. Future work (c) is crucial for these results to become more broadly relevant.

**Questions:**

- Why exactly is it not clear whether or not a Transformer can learn
  unobserved reachability? Are there any computational structures that suggest
  that this might be possible, or any domains with a similar kind of inference
  process where transitivity *was* learned and used?

- What possibility is there for a negative theoretical result, that a
  Transformer of this kind could *never* learn to use transitivity and
  therefore infer reachability?

- In Figure 2, why is $R^{obs}(D_3)$ not identical to $R^{true}$, since $D_3$
  is all possible paths?

**Limitations:**

The authors did not provide a limitations section -- the technical limitations
of the work have been addressed in the weakness section.

---

> ### Author Rebuttal · Authors · 2024-08-05
>
> **Weakness 1**: Need to Improve Logical Connections and Clarity
>
> **Answer**: Thank you for highlighting the need for a clearer introduction and improved logical flow. We will revise the paper accordingly.
>
> Your comment suggests that the inability of LLMs to deduce transitive reachability is reasonable and questions why it is desirable for LLMs to have this capability. We believe this is important because transitivity is a basic property in logic. Since LLMs are often considered close to AGI, it is natural to question whether they can deduce transitivity. Many studies focus on basic properties in logic, such as symmetry [1] and composition [2]. Moreover, we construct a Transformer with perfect performance in path-finding in Theorem 2. This suggests that a Transformer could potentially achieve perfect performance through appropriate training procedures. However, we demonstrate that this is not the case for Transformers trained via next-token prediction.
>
> Regarding the ability to generalize, our analysis shows that no generalizations occur when the embedding size is sufficiently large. A similar phenomenon is reported in [1]. We are unsure if our understanding aligns with your review, particularly the point about the learner's lack of generalization to similar reachable concepts. If you have a different perspective, please feel free to share it with us, and we would be happy to address it during the discussion period.
>
> ---
> **Weakness 2**: Network Architecture Beyond Multi-Layer Perceptrons and Theoretical Justification
>
> **Answer**: In this paper, we aim to study the path-finding task capacity and performance of language models based on the standard Transformer architecture, which is a general-purpose design not specifically intended for graph tasks. Given that the standard Transformer, as specified in the original paper [3] and used in most other LLMs such as Llama-3, employs MLPs in their models, we consider MLPs in our study. While using a GNN instead of an MLP in the Transformer architecture may offer some advantages, it can also introduce several additional challenges, such as requiring prior knowledge of the node's degree, which is beyond the scope of our paper.
>
> Regarding the negative result, we emphasize that we have corresponding theoretical analysis to support this outcome. According to Theorem 3, all unobserved reachability terms $W^V_{j,k}$'s always have a positive gradient, meaning they will continuously decrease during a gradient descent learning procedure. Consequently, unobserved reachability will not be learned in theory. Therefore, "Transformer model has a fundamental difficulty in generating paths for high-degree source-target pairs" is not merely coincidental.
>
> ---
> **Weakness 3**: Experiments on More Domains
>
> **Answer**: We appreciate the feedback on the limitations of our experimental domains. We agree that broader and more realistic datasets are crucial for validating our findings. While our current paper includes a Blocksworld example from PlanBench in Appendix F, we plan to extend our research to encompass a wider range of datasets in future work. This will help us determine whether our findings hold under more complex and realistic scenarios. This line of inquiry, however, is beyond the scope of the current paper and will be summarized into a subsequent paper.
>
> ---
> **Question 1 and Question 2**: Possibility of Learning Unobserved Reachability
>
> **Answer**: This is a very good point. In our future work (b), we mentioned that one important future research topic is to improve the Transformer structure to enable the model to learn unobserved reachability.
>
> The difficulty in learning unobserved reachability with the current Transformer structure arises from the nature of the next-token-prediction loss---learning unobserved reachability results in a higher training loss:
> When predicting the next token with current node $i$ and target node $j$, the distribution of the next token that minimizes training loss  follows the corresponding distribution in the train dataset, i.e., $\Pr[\text{output} = k | \text{current node} = i \text{ and target node} = j] = \frac{N_{i,j,k}}{N_{i,j}}$. If unobserved reachabilities are recorded, they will alter the distribution from $\frac{N_{i,j,k}}{N_{i,j}}$, incurring a higher training loss.
>
> Therefore, we believe that the next-token-prediction loss is one of the reasons the model cannot learn unobserved reachability. With this training loss and current Transformer structure, the model cannot learn to use transitivity as it results in a higher training loss.
>
> However, if another loss is used, such as the accuracy of paths, the Transformer may be able to learn the unobserved reachability. Additionally, if we can improve the Transformer structure to enable the model to "deduce" unobserved reachabilities without recording them, the model may also perform well in the path-finding task.
>
> We will include a discussion on this topic in our final version.
>
> ---
> **Question 3**:  Clarification for $R^{obs}(D_3)$
>
> **Answer**: The distinction between $R^{obs}(D_3)$ and $R^{true}$ arises because $R^{obs}$ includes only the reachability pairs $(j, k)$ where $j$ is a target node and $k$ is a non-source node in the training paths. If $k$ can only be a source node in any path, then reachability $(j, k)$ cannot be included in $R^{obs}$ for any $j$. Consequently, as shown in Figure 2, $R^{obs}(D_3)$ omits all reachabilities involving $k = 0, 1, 4$, since there are no edges pointing to these nodes.
>
>
> ---
> **References**:
>
> [1] Zhu H, Huang B, Zhang S, et al. Towards a Theoretical Understanding of the 'Reversal Curse' via Training Dynamics[J]. arXiv preprint arXiv:2405.04669, 2024.
>
> [2] Yang S, Gribovskaya E, Kassner N, et al. Do Large Language Models Latently Perform Multi-Hop Reasoning?[J]. arXiv preprint arXiv:2402.16837, 2024.
>
> [3] Vaswani A, Shazeer N, Parmar N, et al. Attention is all you need[J]. Advances in neural information processing systems, 2017, 30.

---

> ### Comment · Reviewer_wPpp · 2024-08-09
>
> I thank the authors for their clear rebuttal.
> - Regarding the point about whether your understanding aligns with my review, I think we are on the same page about why generalisation might not happen.
> - Regarding the point about learning unobserved reachability, your answer on the difference between next-token-prediction and predicting the entire path is clear and helpful, but I still wish there had been some evidence included in the paper that unobserved reachability might be learnable at all in this way. It's very good to have a negative result, but what led you to asking the question in the first place?
>
> I am still moderately positively disposed towards this paper, and don't really have any further questions.

---

> ### Author Response · Authors · 2024-08-10
>
> We greatly appreciate your insightful suggestions and comments, which have significantly contributed to the refinement of our paper. We are also thankful for your acknowledgment of our rebuttal efforts. Due to rebuttal space limitations, we cannot explain the rationale for investigating unobserved reachabilities well, which may cause confusion.
>
> **Definition of Observed Reachability:**
> We first revisit the concept of observed reachability, as defined between lines 140 and 154 in our paper. In the context of our experiments, the training dataset is denoted as $\mathcal{D}$. The format for input sequences is 's t s a b c t', where 's' represents the source node, 't' is the target node, and the sequence 's a b c t' constitutes a valid path from 's' to 't'. We define $R^{obs}(t,k)$ as the **observed reachability** from node 'k' to 't', which is determined by the following condition:
> $R^{obs}(t,k) = 1,\text{if } \exists u \in \mathcal{D}, n \in [4,N] \text{ s.t. } u_2 = t, u_n = k$, otherwise $R^{obs}(t,k) = 0$. If node 't' can be reached from 's' while $R^{obs}(t,k)=0$, this is considered an **unobserved reachability**.
>
> **Example:**
> Consider a training dataset containing two sequences: 'a b a b' and 'b d b c d'. The observed reachabilities are (d,c), (b,b), and (d,d). Conversely, the unobserved reachabilities include reachability through transitivity (i.e., (c,a) and (d,a)) and other reachability that does not satisfy the definition (i.e., (b,a), (d,b), (c,c) and (c,b)). For humans, deducing unobserved reachabilities from the given paths is relatively straightforward.
>
> **Rationale for Investigating Unobserved Reachabilities:**
> The motivation for exploring unobserved reachabilities is twofold. Firstly, Algorithm 1 indicates that complete knowledge of all reachabilities is essential for flawlessly completing the path-finding task. Observing the high accuracy of Transformers in path finding, as depicted in Figure 3, leads us to hypothesize that they might infer the true reachabilities. Secondly, Theorem 2 presents a specific configuration of a Transformer's weights that encodes all reachabilities and is capable of finding a path with a high probability. This raises the question: can such weights be derived from the next token prediction loss, a common loss function used by current LLMs?
>
> **Consequences of Inability to Learn Unobserved Reachabilities:**
> The findings presented in Theorem 3 are negative, implying that Transformers with next token prediction loss are unable to infer unobserved reachability through transitivity. Besides the failure of finding a path via transitivity, it has practical implications for compositional reasoning in LLMs. Even if an LLM is aware of the reasoning chains $a\rightarrow b$ and $b\rightarrow c \rightarrow d$, it cannot deduce the extended chain $a \rightarrow b \rightarrow c\rightarrow d$ using transitivity. This limitation applies to current LLMs, including GPT-4, as referenced in [1,2].
>
> [1] Wang B, Yue X, Su Y, et al. Grokked Transformers are Implicit Reasoners: A Mechanistic Journey to the Edge of Generalization[J]. arXiv preprint arXiv:2405.15071, 2024.
>
> [2] Yang S, Gribovskaya E, Kassner N, et al. Do Large Language Models Latently Perform Multi-Hop Reasoning?[J]. arXiv preprint arXiv:2402.16837, 2024.

---

### Official Review · Reviewer_znJo · 2024-07-12

**Soundness:** 3
**Presentation:** 2
**Contribution:** 3
**Rating:** 4
**Confidence:** 4

**Summary:**

The paper investigates planning capabilities in Transformer-based language models by framing planning as a network path-finding task. It reveals that while Transformers can successfully embed adjacency and reachability matrices to perform path-finding, they struggle with transitivity in reachability, limiting their effectiveness in more complex planning scenarios. These theoretical insights are substantiated with experimental validations using both synthetic and real-world datasets, including the Blocksworld benchmark.

**Strengths:**

- The paper addresses a highly significant problem, which is crucial for advancing the field.
- The experimental setup proposed is straightforward yet appears effective, which is commendable.
- The authors have made efforts to approach the topic from both theoretical and empirical perspectives, enriching the study.

**Weaknesses:**

- The paper suffers from a lack of logical connections among the introduction, theory, and experiments. For instance, the broad question posed, "Why does next-word prediction generate intelligence?" lacks a clear alignment with specific aspects of their work. It is unclear which parts of their work address this question and to what extent.
- The clarity of writing needs improvement. For example, the mention of "Project ALPINE" in the introduction is vague, as it does not specify what the project encompasses.

**Questions:**

- Could you clarify whether your project encompasses theory, algorithms, or insights?
- Additionally, it would be beneficial to investigate whether the claims presented in your paper hold up under more realistic setup.

**Limitations:**

- The paper requires a rewrite to better articulate the contributions and clarify the core arguments.
- Further investigation is necessary to assess whether the claims presented hold up under more realistic conditions.

---

> ### Author Rebuttal · Authors · 2024-08-05
>
> We greatly appreciate your valuable input and helpful suggestions. Below, we will address your questions and concerns regarding potential weaknesses in this paper.
>
> ---
>
> **Weakness 1**: About logical connections and the posed broad question: "Why does next-word prediction generate intelligence?"
>
> **Answer**: Thank you for highlighting this issue. Yes, we will address it by improving the introduction to better clarify the structure and connections.
>
> To provide a clearer overview, our research takes a step toward addressing the overarching question: "Why does next-word prediction generate intelligence?" Specifically, we explore this through the lens of planning, a critical component of intelligence, by investigating how next-word prediction facilitates planning, conceptualized as path-finding tasks in unknown networks. This conceptualization is motivated by the planning involved in mathematical proofs, task planning in language agents, and controlled experiments in neuroscience (Lines 43-53).
>
> In this conceptual framework, the language model must effectively learn from path samples of an unknown "ground truth" network to solve path-finding problems. In our work, we conduct both theoretical analyses and targeted experiments to understand the expressiveness and limitations of commonly-used Transformer-based language models in learning to solve path-finding problems from observed path samples. Theoretically, in Section 3.1, we establish that Transformers possess sufficient expressive power to adapt their parameter weights to encode adjacency and reachability matrices. Complementarily, our mathematical analysis of training dynamics in Section 3.2 reveals a fundamental limitation: Transformers trained via next-token prediction can learn adjacency and a limited form of reachability but cannot fully capture reachability through transitivity.
>
> Together, these theoretical analyses establish the following: On one hand, the encoding capabilities of Transformers enable Transformer-based language models to solve path-finding tasks effectively when path samples provide sufficient structural information about the underlying network. On the other hand, due to their limited capacity for transitive inference, commonly used Transformer models may struggle to generalize beyond the observed training data to deduce new reachabilities, unlike human reasoning. A practical implication is composition reasoning: even if LLMs know the reasoning chain $a\rightarrow b \rightarrow c$ and the chain $c\rightarrow d \rightarrow e$, they cannot perform reasoning $a\rightarrow b \rightarrow c \rightarrow d \rightarrow e$ through transitivity. This holds for existing LLMs including GPT-4 [1,2].
>
> In Section 4, we provide targeted experiments to validate these theoretical findings, demonstrating results consistent with our analysis.
>
> While our focus is on the power and limitations of the current "general purpose" Transformer architecture, which is not specifically designed for path-finding tasks, we hope these findings will contribute to designing enhancements for Transformer-based language models and developing learning models tailored for path-finding (and planning) tasks.
>
> ---
>
> **Weakness 2 and Question 1**: About the clarity of writing and what the ALPINE project encompasses.
>
> **Answer**: The project "ALPINE", which stands for **A**utoregressive **L**earning for **P**lanning **I**n **NE**tworks, encompasses conceptualizing planning as path-finding in networks, theoretical analysis on the Transformer structure and auto-regressive loss, and empirical validation of the theoretical analysis. The theoretical analysis also give us some insights about "how the current Transformer do planning", and "what are limitations in the planning capacity of current language models".
>
> We appreciate your suggestion and will clarify these aspects in the final version to ensure a more precise and comprehensive presentation.
>
> ---
>
> **Question 2**: More realistic setup.
>
> **Answer**: We acknowledge the importance of testing our findings in more realistic setups. While our current paper includes an example from Blocksworld from PlanBench in Appendix F, where we use an abstraction to represent all states as unique tokens, we recognize the need for further exploration in more complex and realistic datasets. As mentioned in our future work (c) in Section 5, we plan to extend our research to include such datasets, and aim to investigate whether Transformers can effectively perform abstractions and whether our findings still hold in more realistic scenarios. This line of inquiry, however, is beyond the scope of the current paper and will be summarized into a subsequent paper. Thank you for your suggestion.
>
> ---
>
> **References**:
>
> [1] Wang B, Yue X, Su Y, et al. Grokked Transformers are Implicit Reasoners: A Mechanistic Journey to the Edge of Generalization[J]. arXiv preprint arXiv:2405.15071, 2024.
>
> [2] Yang S, Gribovskaya E, Kassner N, et al. Do Large Language Models Latently Perform Multi-Hop Reasoning?[J]. arXiv preprint arXiv:2402.16837, 2024.

---

> > ### Comment · Reviewer_znJo · 2024-08-12
> >
> > Thanks for your comments. Some of my concerns have been addressed, but I still have some concerns about the paper's writing and the experimental setup, so I keep my score. However, I'm open to the opinions of the other reviewers and AC, and I will respect their final decision.

---

> > > ### Author Response · Authors · 2024-08-13
> > >
> > > We would like to extend our heartfelt thanks for the valuable time and effort you have invested in reviewing our manuscript. Your insightful feedback has significantly contributed to enhancing the manuscript's clarity. We are dedicated to meticulously revising the manuscript in line with your comments.
> > >
> > > Could you please provide additional details regarding your further concerns about the paper's writing style and experimental setup? We welcome any further comments or suggestions you may have that could help improve our paper.

---

### Official Review · Reviewer_T9s7 · 2024-07-13

**Soundness:** 4
**Presentation:** 4
**Contribution:** 3
**Rating:** 7
**Confidence:** 4

**Summary:**

This paper studies the planning capabilities of language models and provides a theoretical foundation for understanding it. The paper investigates the problem by abstracting it as a path-finding problem, showing both theoretically and empirically that transformers can embed adjacency and reachability matrices within their weights. It also highlights their limitations in handling complex planning scenarios.

Main contributions:
- This paper initiates the theoretical study of planning in autoregressive language models by abstracting it as a path-finding problem.
- This paper shows that the transformer has the expressiveness to perform path-finding tasks, and gradient descent on cross-entropy loss cause the Transformer to learn necessary but incomplete graph information for the path-finding task.
- This paper unveils both analytically and empirically that autoregressive training of language models has limitations in the path-finding task.
- The paper analyzes the learning dynamics of a simplified Transformer architecture. It highlights the limitation of transformers to identify reachability relationships through transitivity.
- The theoretical insights are supported by experiments on synthetic path-finding and a real-world planning task (Blocksworld).

The findings contribute to the broader effort of explaining the power and limitations of large language models.

**Strengths:**

Originality: The approach to studying planning capabilities through path-finding in LLMs is novel and insightful. The theoretical aspects are very interesting and inspiring.
Quality: The theoretical studies are well-supported by empirical evidence from synthetic and real-world datasets .
Clarity: The paper is well organized, with clear definitions, methodologies, and analysis of results.
Significance: Understand from a theoretical perspective and conduct empirical studies may help advance its capability in planning.

**Weaknesses:**

- Although the experiments are thorough. They are limited to specific datasets (synthetic and Blocksworld). The paper can benefit from broader validation across diverse planning datasets.
- In my opinion, the practical implications of the theoretical findings are not fully explored. How to leverage the studies still requires further thinking for readers. Further elaboration could strengthen the paper.

**Questions:**

- How do you envision the practical applications of your findings influencing the development of future LLMs, particularly in planning?
- Very interesting results, do you have similar studies for the other benchmark logistics in Planbench, maybe it also worth conducting this study on neutral plan benchmark. https://arxiv.org/abs/2406.04520 (note: not required for this paper, just a suggestion)

**Limitations:**

The authors addressed the limitations in the paper.

---

> ### Author Rebuttal · Authors · 2024-08-05
>
> Thank you for your valuable input. Below, we will address your questions and suggestions concerning potential weaknesses in this paper.
>
> ---
>
> **Weakness 1**: Broader Validation across Diverse Planning Datasets Can be more Beneficial
>
> **Answer**: As mentioned in Section 5, under Future Works (c), we plan to extend our experiments to include a broader range of planning datasets. We hope that this comprehensive expansion of real-world applications will enhance the robustness of our theoretical findings by validating them across more diverse and representative scenarios, while also sharpening our understanding of the limitations of commonly used transformer-based LLM models.
>
> ---
>
> **Weakness 2**: More on Practical Implications Will Strengthen the Paper
>
> **Answer**: While our primary focus was on presenting theoretical analyses aimed at gaining insights into the planning capabilities of language models, we agree that discussing the practical implications is essential. In the final version, we will expand on these aspects; for instance, please refer to the response to Question 1 below.
>
> ---
>
> **Question 1**: Practical Applications in Future LLM Development
>
> **Answer**: Our study demonstrates that the current Transformers can learn observed reachabilities but struggle with unobserved ones.
>
> The latter poses a challenge in planning tasks with limited training data. For instance, given paths "$a$ $c$ $a$ $b$ $c$" and "$c$ $e$ $c$ $d$ $e$" in the training data, the Transformer may not learn how to transition from $a$ to $e$, since it does not know $b$ can reach $e$. This indicates that the existing Transformer architecture is insufficient for achieving human-like planning capabilities. A practical implication is composition reasoning: if LLMs know the reasoning chain from $a\rightarrow b \rightarrow c$ and the chain $c\rightarrow d \rightarrow e$, can they perform reasoning $a\rightarrow b \rightarrow c \rightarrow d \rightarrow e$ through transitivity? The results are consistently negative even for GPT-4 [1,2]. To address this, new Transformer structures that can infer unobserved reachability might be required.
>
> From this perspective, the practical applications of our findings include:
> 1) Understanding how current language models perform planning, i.e., by encoding observed adjacency and reachability in their weights. This can guide us in designing datasets that enable these models to perform well after training.
> 2) By showcasing the limitations of current language models, our study provides a fundamental motivation for future research into models with different Transformer architectures.
> 3) Our simplified demonstrative example serves as a simple yet effective testbed for evaluating new models' capacities in planning, facilitating the development of models that can handle unobserved reachabilities.
>
> ---
>
> **Question 2**: Experiments on Other Benchmark Datasets
>
> **Answer**: We are planning to experiment on other planning related datasets in our next step of research. We believe that by framing planning tasks as path-finding problems and using a suitable abstraction to represent all states as unique tokens---similar to our approach in the Blocksworld example (Appendix F)---the results will likely align with our current findings. Additionally, exploring whether Transformers can perform abstractions on these datasets and understanding how they might do so presents another interesting avenue for future research, which we are currently actively pursuing.
>
> ---
>
> **References**:
>
> [1] Wang B, Yue X, Su Y, et al. Grokked Transformers are Implicit Reasoners: A Mechanistic Journey to the Edge of Generalization[J]. arXiv preprint arXiv:2405.15071, 2024.
>
> [2] Yang S, Gribovskaya E, Kassner N, et al. Do Large Language Models Latently Perform Multi-Hop Reasoning?[J]. arXiv preprint arXiv:2402.16837, 2024.

---

> > ### Comment · Reviewer_T9s7 · 2024-08-12
> >
> > Thank you for your comments and this helped resolve my concerns.

---

### Decision · Program_Chairs · 2024-09-25

**Decision:**

Accept (poster)

**Comment:**

Reviewers agreed the manuscript is interesting and sheds some light on the planning capabilities of Transformers. This is topical as Transformers have not taken over in planning in the way they have in NLP. Work that moves us in the direction of understanding what's different between the two settings and why Transformers are suitable for one case but not for the other is useful for the wider community.

Reviewers would have liked to see if the analysis carries over to some more realistic scenarios, but for an initial paper with this framework they found the setting sufficient. Reviewers also note that this is essentially a negative result but without theory to back it up and make it definitive. Scaling to more scenarios to back up the hypothesis and ultimately finding a theoretical justification for the work would make this much stronger. We can't get there without initial investigations, which is what this is and why it merits acceptance and exposure.